# Intensification of terrestrial carbon cycle related to El Niño–Southern Oscillation under greenhouse warming

Jin-Soo Kim [1], Jong-Seong Kug [1] & Su-Jong Jeong[2]

The El Niño/Southern Oscillation (ENSO) drives interannual variation in the global carbon cycle. However, the relationship between ENSO and the carbon cycle can be modulated by climate change due to anthropogenic forcing. We show herein that the sensitivity of the terrestrial carbon flux to ENSO will be enhanced under greenhouse warming by 44% ($\pm$ 15%), indicating a future amplification of carbon–climate interactions. Separating the contributions of the changes in carbon sensitivity reveals that the response of land surface temperature to ENSO and the sensitivity of gross primary production to local temperature are significantly enhanced under greenhouse warming, thereby amplifying the ENSO–carbon-cycle coupling. In a warm climate, depletion of soil moisture increases temperature response in a given ENSO event. These findings suggest that the ENSO-related carbon cycle will be enhanced by hydroclimate changes caused by anthropogenic forcing.

[1] Division of Environmental Science and Engineering, Pohang University of Science and Technology (POSTECH), Pohang 37673, South Korea. [2] School of Environmental Science and Engineering, Southern University of Science and Technology (SUSTECH), Shenzhen 518055, China. Correspondence and requests for materials should be addressed to J.-S.K. (email: jskug@postech.ac.kr) or to S.-J.J. (email: sujong@sustc.edu.cn)

The El Niño/Southern Oscillation (ENSO) strongly impacts the interannual variations in the global carbon cycle by influencing terrestrial ecosystem processes via extensive teleconnection[1]. Throughout most of the tropics, terrestrial vegetation productivity is reduced by the increased surface temperature and decreased precipitation associated with El Niño, whereas the opposite anomalies occur in the case of La Niña[2–6]. This response of terrestrial ecosystems to ENSO controls significant changes in terrestrial productivity and the magnitude of the terrestrial carbon flux, eventually regulating the concentration of atmospheric $CO_2$[7, 8]. Therefore, understanding the relation between ENSO and the terrestrial carbon cycle provides an excellent route to predicting changes in the global carbon cycle.

In a warmer climate, the terrestrial carbon cycle will change in response to altered ENSO teleconnection[9]. For example, previous studies indicate that greenhouse warming would cause changes in the regional impacts of ENSO[10–14] because ENSO responses can be altered by changes in the mean background states, such as atmospheric circulation patterns, surface temperature, and precipitation. Changes in these atmospheric conditions associated with ENSO could also affect their impacts on the terrestrial

ecosystem and carbon cycle. In addition, regional changes in the mean climate lead to changes in the sensitivity of the terrestrial carbon flux to local temperature anomalies[15–17], which means that changes in sensitivity depending on varying climate states will be directly related to the strength of the terrestrial response to ENSO. Greenhouse warming leads to a strong potential for a change in the relation between ENSO and the terrestrial carbon cycle, thereby requiring quantitative investigations to estimate future changes in the ENSO-related carbon cycle.

In the present study, we examine future changes in the sensitivity of the terrestrial carbon cycle to ENSO in the Coupled Model Intercomparison Project Phase 5 (CMIP5) Earth System Models (ESMs) by comparing preindustrial and future projections. We analyze the Extended Concentration Pathway 4.5 (ECP4.5) scenario for the future projections, which extends the Representative Concentration Pathways (RCP) 4.5 with continuous anthropogenic forcing until 2300. It is difficult to estimate temporal changes of interannual sensitivities using the RCP scenario because atmospheric conditions and carbon fluxes are strongly affected by anthropogenic forcing with considerable year-to-year variations and a time-varying trend with respect to

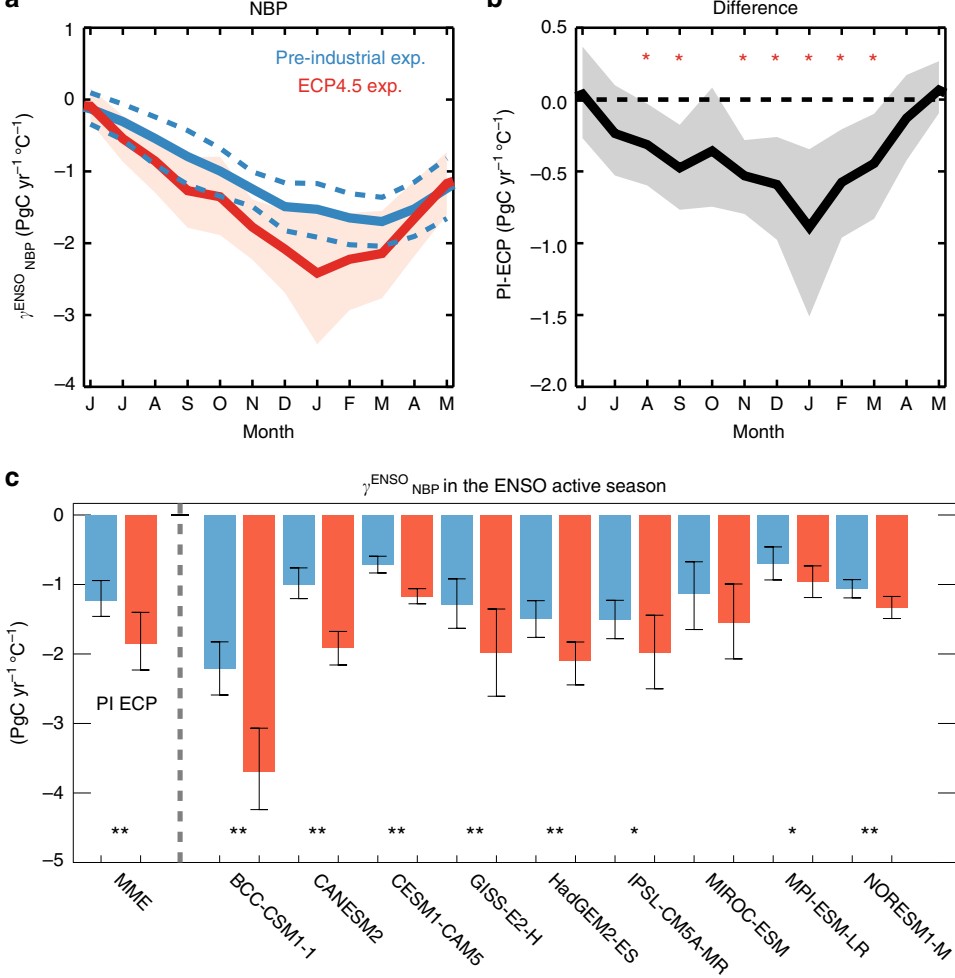

**Fig. 1** Net biome production sensitivities to ENSO. **a** Regression coefficients of carbon-flux anomalies in the range 40 °S–40 °N for the December–February (DJF) Niño3.4 index for net biome production (NBP) $\left(\gamma_{\text{NBP}}^{\text{ENSO}}\right)$ based on the preindustrial experiment (blue) and ECP4.5 (red), and **b** the differences between the two experiments $\left(\Delta\gamma_{\text{NBP}}^{\text{ENSO}}\right)$. Shaded area and dashed lines indicate 95% confidence levels calculated using the bootstrap method. Red stars indicate significant months for $\Delta\gamma_{\text{NBP}}^{\text{ENSO}}$ at 95% confidence levels. **c** Regression coefficients of NBP anomalies from September to February in the range 40 °S–40 °N for DJF Niño3.4 for the preindustrial experiment and ECP4.5 based on multimodel ensemble (MME, left) as simple arithmetic mean of nine Earth System Models (ESMs) and individual ESMs (right). Error bars indicate 95% confidence levels of regression coefficients calculated by using bootstrap method. *P < 0.1 and **P < 0.05 for difference between two experiments in regression coefficients being significantly different from zero

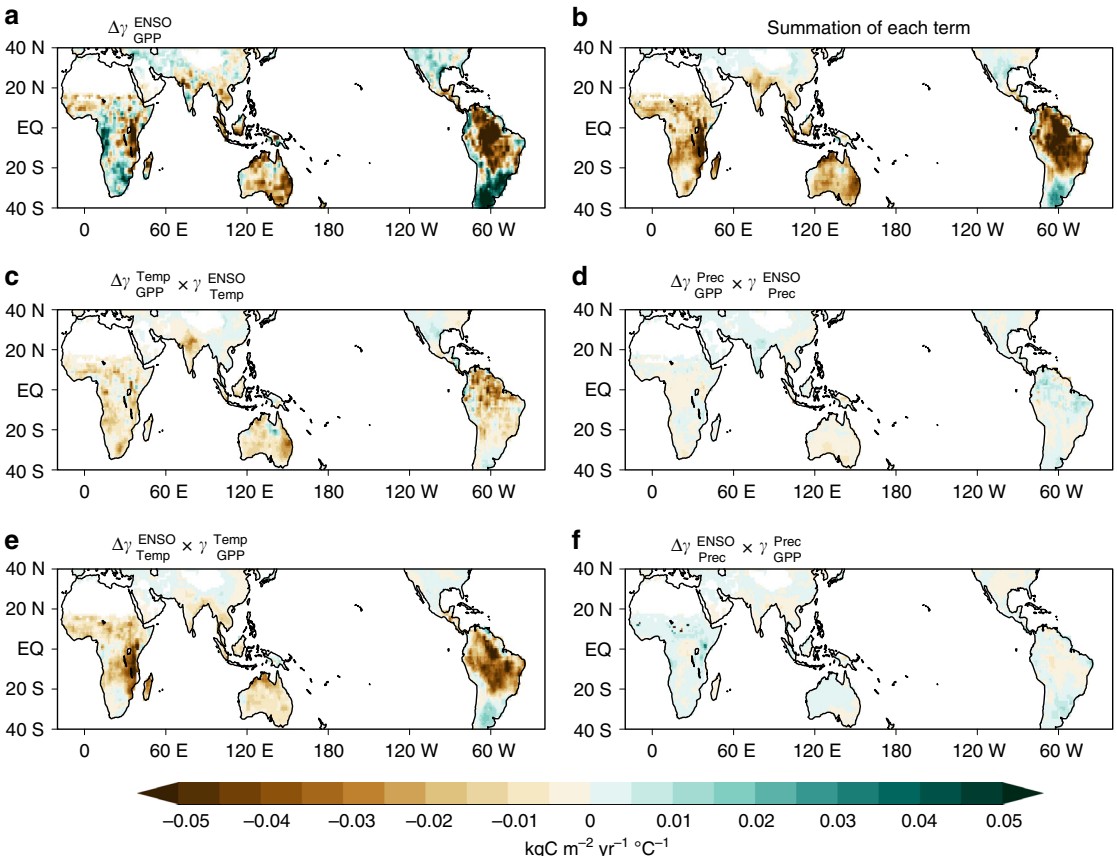

**Fig. 2** Sensitivity contributions to enhanced gross primary production response to ENSO. **a** Differences in the regression coefficient of gross primary production (GPP) anomalies $\left(\Delta\gamma_{\mathrm{GPP}}^{\mathrm{ENSO}}\right)$ from September to February for the December–February Niño3.4 index between the preindustrial experiment and ECP4.5. **b** Sum of changes in sensitivities based on each term in Eq. (4). **c–f** Each term in Eq. (4) plotted as a contribution to future changes in the response to ENSO and the GPP sensitivity changes to **c** temperature $\left(\Delta\gamma_{\mathrm{GPP}}^{\mathrm{Temp}}\right)$ and **d** precipitation $\left(\Delta\gamma_{\mathrm{GPP}}^{\mathrm{Prec}}\right)$; the GPP response to ENSO due to **e** temperature $\left(\Delta\gamma_{\mathrm{Temp}}^{\mathrm{ENSO}}\right)$ and **f** precipitation $\left(\Delta\gamma_{\mathrm{Prec}}^{\mathrm{ENSO}}\right)$. Units are kgC m$^{-2}$ yr$^{-1}$ °C$^{-1}$

the ongoing climate change. To estimate more stable interannual sensitivities under the influence of anthropogenic climate change, we use a stabilized period of 200 years of ECP4.5, which is an idealized experiment for the 22nd and 23rd centuries. In this way, interannual sensitivities are more statistically significant and thus better estimate the anthropogenic effects on these sensitivities, allowing for a better comparison with sensitivities obtained from the preindustrial experiment (see Methods).

## Results

**Enhanced ENSO-related carbon cycle under greenhouse warming.** The sensitivity to ENSO ($\gamma_{\mathrm{NBP}}^{\mathrm{ENSO}}$; PgC yr$^{-1}$ °C$^{-1}$) of net carbon flux from land (net biome production; NBP) in the range 40°S–40°N is estimated based on a linear regression of NBP anomalies with respect to December–February (DJF) Niño3.4 index. This sensitivity is well simulated by the CMIP5 ESMs and is consistent with previous modeling results in terms of phase and peak timing[18, 19] (Fig. 1a). As shown in Fig. 1, the ESMs show negative values, suggesting that the ESMs capture well the decreased carbon uptake during the El Niño years and the increased carbon uptake during the La Niña years (see Supplementary Fig. 1). The maximum NBP response to ENSO also appears the following March, whereas the ENSO magnitude usually peaks in the boreal winter[4, 8, 18, 19]. This delayed peak in the ENSO-related carbon flux is consistent with the findings of numerous studies based on both observation and modeling[18–21].

It is evident that the sensitivity is largely increased in the ECP4.5 simulation (Fig. 1a), accompanied by a stronger sensitivity of NBP to ENSO in the projection. This result suggests that ENSO-related NBP intensifies under greenhouse warming. This intensification $\left(\Delta\gamma_{\mathrm{NBP}}^{\mathrm{ENSO}}\right)$ appears clearly from September to February (SONDJF; Fig. 1b, 44.9% for SON and 44.1% for DJF), when ENSO anomalies are strong. The NBP response to ENSO is −1.29 PgC yr$^{-1}$ °C$^{-1}$ in the preindustrial experiment for the SONDJF mean; however, it is enhanced by 44.4% in the future projection. The increase in the multimodel ensemble (MME) is also significant at the 95% confidence level, based on bootstrap estimates. In addition to the MME, all individual models indicate that compared with the results of the preindustrial experiment, the NBP will become more sensitive to ENSO under greenhouse warming (Fig. 1c). In particular, among nine ESMs, six (eight) models exhibit significantly stronger sensitivities of NBP in the period SONDJF at the 95% (90%) confidence level.

The NBP carbon flux is defined as the sum of carbon fluxes due to gross primary production (total biomass produced by photosynthesis; GPP), autotrophic respiration $R_a$, heterotrophic respiration $R_h$, and emissions from wildfires in the CMIP5 ESMs. The sensitivity of carbon fluxes to ENSO is estimated from the separate responses of each carbon flux. The GPP response to ENSO, $\gamma_{\mathrm{GPP}}^{\mathrm{ENSO}}$, explains most of the NBP response to ENSO in the ESMs, but carbon fluxes due to respiration, $R_a$ and $R_h$, are relatively small for both the experiments and for the differences between the two experiments (see Supplementary Fig. 2d, f).

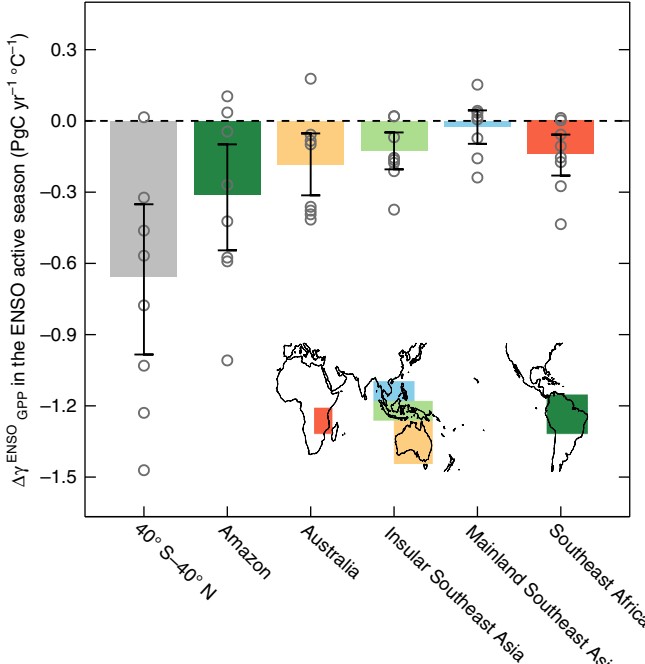

**Fig. 3** Regional contributions of enhanced gross primary production response to ENSO. Differences in the regression coefficients of regional gross primary production anomalies from September to February for the December–February Niño3.4 index between the preindustrial experiment and ECP4.5. Each bar shows the results of the multimodel ensemble based on each region as marked on the map. Open circles denote the individual results of the ESM. Error bars indicate 95% confidence levels calculated using the bootstrap method

Because the CMIP5 models tend to underestimate these ENSO-associated fluxes in the control experiment[19], changes in respiration will require further examination. However, projections of carbon flux due to wildfires change significantly as compared to preindustrial experiments, which give −0.12 PgC yr$^{-1}$ in SONDJF (see Supplementary Fig. 2h). This result is consistent with a previous study that shows enhanced fire activity associated with El Niño due to water deficit under a warm climate[22]. Although this MME result is about 20% of GPP anomalies in SONDJF, it is based on only four models: CESM1-CAM5, IPSL-CM5A-MR, MPI-ESM-LR, and NORESM1-M, which simulate carbon flux due to wildfires. Because other models do not include carbon-emission anomalies due to fire, the CMIP5 ESMs tend to underestimate the contribution of fires to NBP anomalies. Nonetheless, the individual ESMs and the MME mostly exhibit similar magnitudes, evolutions, and peak timings in the GPP response to ENSO as compared with the NBP response to ENSO (see Supplementary Fig. 3). Thus, the carbon fluxes according to GPP anomalies in the range 40 °S–40 °N mainly lead to terrestrial carbon-flux anomalies associated with ENSO in the ESMs.

**Contributions to enhanced sensitivity**. Because the terrestrial GPP is significantly affected by local temperature and precipitation (see Methods, Eq. (2))[7, 8, 18, 19], the GPP response to ENSO is separated into the contributions of local temperature and precipitation to understand how the ENSO-related carbon cycle intensifies under greenhouse warming. The future changes in the GPP response to ENSO are determined by changes in the response of local temperature and precipitation to ENSO and changes in the sensitivity of GPP to local temperature and precipitation (see

Methods, Eq. (4)). Figure 2 shows the contribution of each term to the increased GPP responses to ENSO under greenhouse warming. In Fig. 2b, the sum of the linear terms is similar in spatial distribution and magnitude to the simulated changes in the GPP response (see Fig. 2a and Supplementary Fig. 4), suggesting that this separation is valid. For western Africa, however, the sum of all terms cannot explain the negative value of the changes in the GPP response to ENSO, suggesting that other physical factors might be more important for changes over western Africa.

For most tropical regions, such as in Amazonia, Australia, and Insular Southeast Asia, we find a negative contribution of the changes in the GPP sensitivity to temperature ($\Delta\gamma_{\mathrm{GPP}}^{\mathrm{Temp}}\gamma_{\mathrm{Temp}}^{\mathrm{ENSO}}$, Fig. 2c and Supplementary Fig. 5) and in the temperature response to ENSO ($\Delta\gamma_{\mathrm{Temp}}^{\mathrm{ENSO}}\gamma_{\mathrm{GPP}}^{\mathrm{Temp}}$, Fig. 2e and Supplementary Fig. 6). This indicates that these changes significantly contribute to the changes in the GPP response, whereas the other parameters make relatively small contributions related to precipitation[23]. As shown in Fig. 2c, the changes in the GPP sensitivity to temperature $\left(\Delta\gamma_{\mathrm{GPP}}^{\mathrm{Temp}}\gamma_{\mathrm{Temp}}^{\mathrm{ENSO}}\right)$ also substantially contribute throughout the tropics to the changes in the enhanced GPP response to ENSO $\left(\Delta\gamma_{\mathrm{GPP}}^{\mathrm{ENSO}}\right)$ (Fig. 2c). This result suggests that the carbon flux of GPP will be more sensitive to local temperature variations under greenhouse warming, which is consistent with previous studies that suggest that the current climate is already close to or exceeding the optimal temperature range for terrestrial productivity[16, 17]. Thus, even if the remote impacts of ENSO on regional climate are not altered under greenhouse warming, the ENSO-related carbon cycle can be intensified because of changes in GPP sensitivity to temperature $\left(\Delta\gamma_{\mathrm{GPP}}^{\mathrm{Temp}}\right)$.

In addition, Fig. 2e $\left(\Delta\gamma_{\mathrm{Temp}}^{\mathrm{ENSO}}\gamma_{\mathrm{GPP}}^{\mathrm{Temp}}\right)$ shows the contribution of changes in the response of land temperature to ENSO $\left(\gamma_{\mathrm{Temp}}^{\mathrm{ENSO}}\right)$ on changes in the GPP response to ENSO. Because the GPP response to local temperature $\left(\gamma_{\mathrm{GPP}}^{\mathrm{Temp}}\right)$ is mostly negative in tropical regions, enhanced $\gamma_{\mathrm{Temp}}^{\mathrm{ENSO}}$ under greenhouse warming leads to a negative GPP response to ENSO. In particular, the enhanced temperature response to ENSO clearly appears over the Amazon and eastern Africa. The enhanced temperature response during El Niño tends to reduce the carbon uptake by GPP owing to enhanced heat stress to ecosystems.

**Regional contributions in the future changes**. The stronger terrestrial carbon-flux anomalies associated with ENSO in a warm climate appear in most tropical regions; however, their contributions vary over different regions. To examine which regions contribute the most to the stronger GPP response to ENSO, we separate changes in the GPP response to ENSO into six land regions in the tropics and the subtropics that are known to have dominant terrestrial carbon fluxes. Of these regions, Amazonia, Australia, Insular Southeast Asia, and Southeast Africa considerably contribute to the reduced carbon fluxes out of atmosphere due to GPP on land (i.e., −0.31, −0.18, −0.13, and −0.14 PgC yr$^{-1}$ °C$^{-1}$, respectively; see Fig. 3). These regional contributions appear to be proportional to their present contributions, except for Southeast Africa (see Supplementary Fig. 7). Because almost half of the changes in the GPP response to ENSO can be attributed to the contribution of the Amazon region, the Amazon terrestrial carbon flux changes are the most critical for changes in the interannual variability of the global carbon cycle, as suggested by previous studies[24, 25]. Moreover, the increase in Australia is consistent with the recent intensified interannual carbon flux variability in semi-arid regions, which would be an important driver of the interannual global carbon cycle in the future[26]. Both

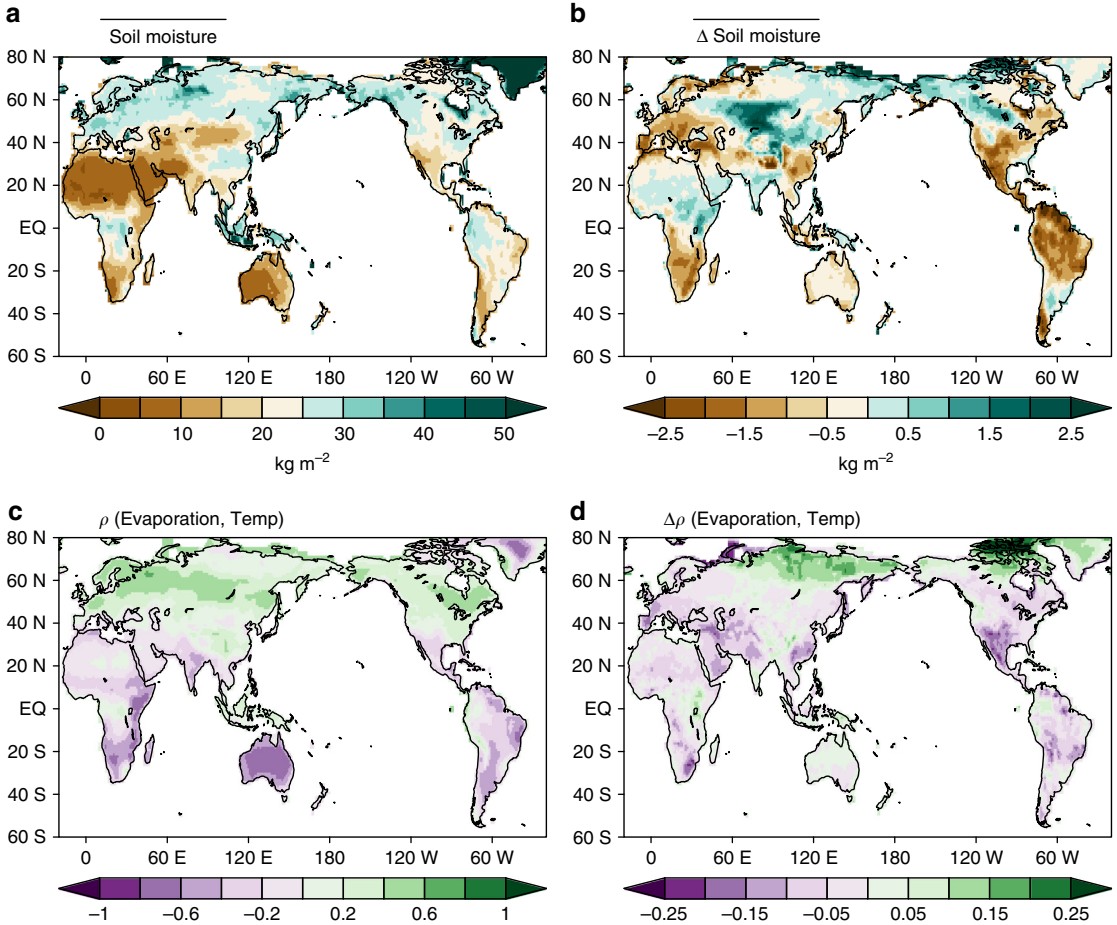

**Fig. 4** Soil moisture and evaporation-temperature coupling. **a** Climatological soil-moisture content from September to February in the preindustrial experiment (kg m$^{-2}$) and **b** soil-moisture climatology differences between the preindustrial experiment and ECP4.5 (kg m$^{-2}$). **c** Correlation coefficients of evaporation at the local surface temperature in the preindustrial experiment, **d** and the difference in the correlation coefficients between the preindustrial experiment and ECP4.5

changes in GPP response to ENSO $\left(\Delta\gamma_{GPP}^{ENSO}\right)$ in Amazonia and Australia can be explained by $\Delta\gamma_{Temp}^{ENSO}\gamma_{GPP}^{Temp}$ and $\Delta\gamma_{GPP}^{Temp}\gamma_{Temp}^{ENSO}$, but $\Delta\gamma_{Temp}^{ENSO}\gamma_{GPP}^{Temp}$ is the most dominant contributor for the Amazon region (see Supplementary Fig. 8). However, the current models also show a considerable spread in future changes, indicating large uncertainty. This uncertainty reveals that current land models have large uncertainties regarding ENSO teleconnection[27] and terrestrial production to climatic conditions[28]. Despite the wide model spread, the conclusion remains robust: changes in the GPP response to ENSO in the tropics lead to global changes in the GPP response to ENSO.

**Possible mechanism for the enhanced temperature responses.**
The enhanced land temperature responses to ENSO $\left(\Delta\gamma_{Temp}^{ENSO}\right)$ substantially contribute to the changes in the GPP response to ENSO (Fig. 2 and Supplementary Figs. 6 and 9); however, it remains unclear how the land temperature responses become stronger during the ENSO phases. A previous modeling study suggests that this pattern may be attributed to surface heat budget changes caused by cloudiness and decreased evaporation related to soil-moisture depletion[10]. However, cloudiness and its relevant surface heat budget do not show significant changes at least in the nine ESMs (see Supplementary Fig. 10). In contrast, previous studies suggest that soil-moisture depletion under greenhouse

warming can increase the possibility of extreme heat events not only in Europe and North America but also in the tropics[29–32]. The soil moisture in the preindustrial experiment is greater than that in the future projection across the entire tropics, except for western Africa and India (Fig. 4b), and this result is consistent with that of a previous multimodel study[33]. Soil-moisture depletion leads to enhanced sensible heat exchange between the land surface and air rather than latent heat flux, and this is particularly effective in areas with a soil-moisture-limited regime[30].

Figure 4c shows the correlation $\rho(E, T)$ between evaporation and surface temperature, which is a useful diagnostic tool for estimating the coupling strength between evaporation and temperature[31]. The ESMs exhibit a strong negative relation between evaporation and surface temperature over soil-moisture-limited regions in the tropics, including Amazonia, Australia, South Asia, and Africa (Fig. 4a, c). This negative correlation means that decreased evaporation is closely related to an increase in the local temperature anomaly through enhanced sensible heat flux exchange, regardless of latent cooling for those regions, and suggests a soil-moisture-limited regime[29–36]. As expected, the spatial pattern of future changes in the evaporation-temperature coupling strength, $\Delta\rho(E, T)$, is fairly similar to that of the future changes in soil moisture (Fig. 4b, d). This implies that the soil-moisture-limited regime will expand in the tropics under greenhouse warming and that evaporation-temperature coupling

will be enhanced accordingly. Therefore, future strengthened evaporation-temperature coupling indicates that the initial temperature anomalies may be intensified further and maintained longer as a result of the strong positive feedback of soil moisture. Consequently, the response of the ENSO-induced land temperature is also enhanced as the evaporation-temperature coupling strengthens. In other words, reduced soil moisture under greenhouse warming leads to enhanced ENSO-related temperatures owing to the strengthened evaporation-temperature coupling (see Supplementary Fig. 11).

## Discussion

A comparison of two different scenarios (preindustrial & ECP4.5) in the ESM simulations shows that the sensitivity of terrestrial carbon flux to ENSO is enhanced under greenhouse warming. The amplification of the coupling between ENSO and the global carbon cycle results from two physical processes: (i) the enhanced sensitivity of GPP to local temperature and (ii) the enhanced ENSO impacts on the land temperature. These two factors are expected to affect other interannual climate variabilities and long-term climate change. In other words, the results of the present study suggest that two-way coupling between the climate system and the global carbon cycle might be intensified in a warmer climate; therefore, climate sensitivity to anthropogenic forcing may accelerate as global warming progresses.

In contrast to GPP, $R_a$ and $R_h$ show no significant changes between the two scenarios (see Supplementary Fig. 2). Indeed, attempting various parameterizations for $R_h$ related to temperature variation in the CMIP5 ESMs[37] shows that a systematic problem exists in the current land surface models for capturing the temperature sensitivity of respiration, especially that linked to long-term temperature variability[38]. To estimate future changes in the $R_h$ response to ENSO, the model parameterization for $R_h$ should be further validated and investigated[39]. In addition, current ESMs, including process-based fire schemes, tend to underestimate the effects of wildfires[19, 40, 41], which implies that our results may have overlooked the contributions of changes in $R_h$ and changes due to wildfires related to $\Delta\gamma_{\text{Temp}}^{\text{ENSO}}$. Such changes might further intensify the ENSO-related terrestrial carbon flux, rather than what we report herein.

In addition, our results imply that the amplified ENSO-related interannual carbon cycle can change the terrestrial productivity on a long-term time scale, particularly in regions that are vulnerable in terms of climatic extremes. Observational studies have pointed out that climate extremes (e.g., heat waves) will even cause collective mortality in tropical forests by exceeding the threshold for survivable conditions[42–45]. Consequently, the damaged terrestrial ecosystem will partly lose its function of consuming $CO_2$ until natural restoration. Thus, short-term climate extremes could cause a considerable reduction in long-term carbon uptake and may promote positive feedback to greenhouse warming[46–48]. For example, numerous studies have warned that land use change, deforestation, and dieback of the Amazon forest will accelerate future global warming[22, 49–52]. Indeed, stronger ENSO impacts on Amazonian terrestrial production would promote accelerated $CO_2$-induced global warming through extensive dieback.

Furthermore, in addition to regions of tropical rainforest, the present results show that regions with high cropland intensity, such as South Asia and East Australia, exhibit enhanced GPP response to ENSO under greenhouse warming. These regions considerably contribute to global crop production and a close relation exists between ENSO and crop production in these regions[13, 53]. Therefore, our results imply that enhanced ENSO impacts on terrestrial productivity will lead to larger interannual variability in the global crop yield associated with ENSO. Although a strong positive trend exists in global crop production based on technical progress in agriculture, approximately one-third of the global crop yield can be explained by climate variability[54], which means that enhanced ENSO impacts on terrestrial response play a role in both the global carbon cycle and food security because they influence global crop production.

## Methods

**Data.** Nine ESMs from CMIP5 were analyzed (BCC-CSM1-1, CANESM2, CESM1-CAM5, GISS-E2-H, HADGEM2-ES, IPSL-CM5A-MR, MIROC-ESM, MPI-ESM-LR, and NORESM1-M) based on the availability of carbon fluxes and long-term simulations. We used the preindustrial experiment and ECP4.5[55] to estimate the future changes in ENSO teleconnection and its relevant carbon-cycle variations (see Supplementary Table 1). The MME for the regression coefficient is given by the simple arithmetic mean of nine ESMs. Bootstrap resampling is used to assess the significance of MME. By randomly selecting nine ESMs with replacements, 10,000 bootstrap samples are produced for the regression coefficient. These are used to produce 10,000 estimates of the MME, which constitute an empirical bootstrap distribution and provide a confidence interval for the MME. ECP4.5 is the extension of RCP4.5 with continuous anthropogenic forcing until 2300. Previous studies attempted to estimate the anthropogenic effects on the climate by comparing climatology in the first (i.e., 2006–2025) and the last (i.e., 2081–2100) period of the 21st century projection. However, estimating the interannual sensitivities is difficult even with a 20- or 30-year window because variables are strongly affected by anthropogenic forcing with considerable year-to-year variations and a time-varying trend with respect to the ongoing climate change[56–58]. Moreover, the long-term natural variability inherent in model simulations can hinder the estimation of accurate interannual sensitivities when using a 20- or 30-year window[59]. To more accurately estimate sensitivities under the influence of anthropogenic climate change, we use a stabilized period of 200 years of ECP4.5 (i.e., the 22nd and 23rd centuries). In this way, interannual sensitivities have statistical significance for estimating the anthropogenic effects on these sensitivities when comparing the sensitivities obtained using the preindustrial experiment.

The CMIP5 ESMs provide carbon-flux variables between land and atmosphere related to land processes such as terrestrial production and respiration. In this study, carbon fluxes are defined as the amount of $CO_2$ transported from land to the atmosphere; thus, a negative value means that land emits $CO_2$ into the atmosphere and a positive value implies that land removes $CO_2$ from the atmosphere.

**Sensitivity analysis.** The DJF Niño3.4 (5 °S–5 °N, 170°–120 °W) index is used to represent the ENSO state (4), and linear regression is used to estimate the impact of ENSO on regional temperature, precipitation, and carbon fluxes. We define these regression coefficients as sensitivity ($\gamma$). For example, the temperature sensitivity to ENSO ($\gamma_{\text{Temp}}^{\text{ENSO}}$) is defined as the regression coefficient between the DJF Niño3.4 index and temperature anomalies. Similarly, $\gamma_{\text{Prec}}^{\text{ENSO}}$ indicates a regression coefficient that falls between the DJF Niño3.4 index and the precipitation anomalies. Future change in the sensitivity $\Delta\gamma$ is defined as the difference in sensitivity between preindustrial and ECP4.5 simulations. The MME was calculated by averaging the regressed results from each of the nine ESMs. Model outputs were regridded to a common 1° × 1° latitude-longitude grid to obtain Figs. 3 and 4.

Surface temperature and precipitation are highly correlated to each other over the tropics and their effects on the carbon-flux anomalies cannot be easily distinguished based on a simple linear regression[19]. Thus, to separately estimate the sensitivities of GPP to temperature and precipitation, a partial regression was used herein on behalf of the partial differential for the sensitivity of the GPP to surface temperature and precipitation, and partial regression coefficients are derived from multiple linear regressions of GPP on surface temperature and precipitation[17, 19, 60, 61]. This may be expressed as

$$\frac{\partial(\text{GPP})}{\partial(\text{Temp})}\delta\text{Temp} + \frac{\partial(\text{GPP})}{\partial(\text{Prec})}\delta\text{Prec} + \varepsilon$$
$$= \gamma_{\text{GPP}}^{\text{Temp}}\delta\text{Temp} + \gamma_{\text{GPP}}^{\text{Prec}}\delta\text{Prec} + \varepsilon, \qquad (1)$$

where $\gamma_{\text{GPP}}^{\text{Temp}}$ and $\gamma_{\text{GPP}}^{\text{Prec}}$ are obtained from the coefficients based on the partial regression method. These parameters approximately represent the sensitivities of GPP to surface temperature and precipitation, respectively. Moreover, to quantify the changes associated with ENSO forcing, GPP changes with respect to ENSO forcing may be expressed as

$$\frac{\text{d}(\text{GPP})}{\text{d}(\text{ENSO})} = \frac{\partial(\text{GPP})}{\partial(\text{Temp})}\frac{\text{d}(\text{Temp})}{\text{d}(\text{ENSO})} + \frac{\partial(\text{GPP})}{\partial(\text{Prec})}\frac{\text{d}(\text{Prec})}{\text{d}(\text{ENSO})} + \varepsilon$$
$$= \gamma_{\text{GPP}}^{\text{Temp}}\frac{\text{d}(\text{Temp})}{\text{d}(\text{ENSO})} + \gamma_{\text{GPP}}^{\text{Prec}}\frac{\text{d}(\text{Prec})}{\text{d}(\text{ENSO})} + \varepsilon \qquad (2)$$

where the ordinary differential is used for the sensitivity of temperature and precipitation to ENSO forcing. The partial differential is applied for the sensitivity of the GPP to temperature ($\gamma_{\text{GPP}}^{\text{Temp}}$) and precipitation ($\gamma_{\text{GPP}}^{\text{Prec}}$). We define each term

in Eq. (2) as sensitivity, which may be expressed in a simplified form as

$$\gamma_{\text{GPP}}^{\text{ENSO}} = \gamma_{\text{GPP}}^{\text{Temp}}\gamma_{\text{Temp}}^{\text{ENSO}} + \gamma_{\text{GPP}}^{\text{Prec}}\gamma_{\text{Prec}}^{\text{ENSO}} + \varepsilon \tag{3}$$

Based on Eq. (3), future changes in GPP responses to the ENSO forcing $\left(\Delta\gamma_{\text{GPP}}^{\text{ENSO}}\right)$ can be separated by four terms if nonlinear terms are ignored. This can be expressed as

$$\Delta\gamma_{\text{GPP}}^{\text{ENSO}} = \Delta\gamma_{\text{GPP}}^{\text{Temp}}\gamma_{\text{Temp}}^{\text{ENSO}} + \Delta\gamma_{\text{Temp}}^{\text{ENSO}}\gamma_{\text{GPP}}^{\text{Temp}} + \Delta\gamma_{\text{GPP}}^{\text{Prec}}\gamma_{\text{Prec}}^{\text{ENSO}} + \Delta\gamma_{\text{Prec}}^{\text{ENSO}}\gamma_{\text{GPP}}^{\text{Prec}} + \varepsilon. \tag{4}$$

Each term in Eq. (4) indicates contributions to $\Delta\gamma_{\text{GPP}}^{\text{ENSO}}$. The spatial patterns of these four terms are shown in Fig. 3 to investigate their relative contributions to the regional aspects.

**Data availability**. The data that support the findings of this study are all publicly available from their sources. Processed data, products, and code produced in this study are available from the corresponding author upon reasonable request.

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

## Acknowledgements

We acknowledge the World Climate Research Programme's Working Group on Coupled Modelling, which is responsible for CMIP, and the climate modelling groups (listed in Supplementary Table 1) for producing and making available their model output. J.-S. Kug and J.-S. Kim were supported by the National Research Foundation of Korea (NRF-2017R1A2B3011511). S.-J.J. was supported by the Southern University of Science and Technology (No. G01296001).

## Author contributions

J.-S. Kim compiled the data, conducted analyses, prepared the figures, and wrote the manuscript. J.-S. Kug and S.-J.J. designed the research and wrote majority of the manuscript content. All of the authors discussed the study results and reviewed the manuscript.

## Additional information

**Competing interests:** The authors declare no competing financial interests.

**Change history:** A correction to this article has been published and is linked from the HTML version of this paper.

