## [Peer Review File · Nature Communications]

Reviewer #1 (Remarks to the Author):

This is an important and interesting subject, and despite my generally critical comments to follow, I think this is excellent work and worthy of publication after some additional work. This topic has not been addressed in a significant way since Wang and Schimel's 2003 review, but as evidence grows for the powerful effect of El Nino on tropical carbon cycling, it is of growing concern relative to the future.

My concern is as follows. The authors do a thorough analysis of the CMIP ensemble but don't address either its significant divergence, suggesting fundamental disagreement over mechanism nor comparison with observations.

This raises two issues that have to be dealt with. First, in contrast to physical ensembles, there is little evidence that the central tendency of the carbon ensemble is more skillful than the individual models. In fact, model evaluations show the ensemble to be significantly biased relative to observational constraints (papers by Hoffman, Cox and Friedlingstein). While these results have challenges of their own, they suggest that use of the ensemble is not as well justified as for physical climate.

Second, the authors focus in on GPP as the primary driver of response to El Nino. This is also not justified relative to observations, and calls the exercise into question. The literature demonstrates conclusively that a large part of the El Nino cycle is due to variation in fire, and this both affects NEE directly, and should influence it in a lagged fashion, as burned areas regrow. Omitting this is not justified (papers by Francey, van der Werf, Bloom and many others). Second, some studies (papers by Anderegg and Clark) suggest a prominent role for respiration in tropical El Nino responses. Taken together, this conflicts with the model's primary response occurring through GPP and suggests the models, even when they replicate observed patterns are doing so for the wrong reasons. Forthcoming papers from the OCO-2 science team, as presented at AGU last year, will show even greater complexity (poster presentations by Liu, Chatterjee and others).

As a result of these two concerns, in contrast to current diagnostic papers on the physical system, where a certain measure of skill can be taken for granted and is established in the literature, the carbon cycle models cannot be assumed to have similar levels of skill, and this analysis itself suggests they may all respond to El Nino-type variability incorrectly.

A diagnostic paper on the response of the CMIP ensemble to El Nino variability is of great interest and value, but drawing scientific conclusions without dealing with the potentially near-total lack of correct mechanism is unwarranted. It may well be, indeed, I expect that the authors conclusions about carbon-water coupling remain robust to these model flaws, but the paper is misleading if not presented in the perspective of how immature and erroneous these models actually are.

Reviewer #2 (Remarks to the Author):

Kim et al. report the results of an assessment of the sensitivity of the carbon cycle to the El Nino Southern Oscillation using output from the CMIP5 models, both under current climate conditions and future climate change. They show a consistent response of the CMIP5 models in terms of the sensitivity to ENSO events, and the seasonal distribution. More importantly, they show an increase in the sensitivity under future climate change. The paper is well written, the analysis and methodology well described, and the results will no doubt be of interest to the readers of Nature Communications.

General comments:

It is not clear how the sensitivity metric is defined for NBP. For ENSO-Temp relationships it is defined based on anomalies, but the definition for NBP is never given, other than saying 'we estimated the sensitivity of net biome production (NBP) to ENSO on the basis of the linear regression in the ESMs.' Are NBP anomalies used, and normalized units? Could the sensitivity increase just because NBP increases?

Why did the authors use RCP4.5? Presumably RCP8.5 would give clearer results? It would be informative to compare results from the two scenarios.

Detailed comments:

The sentences from Line 54 to 61 repeat the same statement in different ways.

Line 61: It is unclear why mean climate states would change the sensitivity. Are the authors suggesting that the response is a non-linear function of mean climate state?

Line 72: 'as well as' 200 years of ECP4.5

Line 78: 'linear regression in the ESMs'. It is not clear what you mean here. I presume the linear regression of the relationship between NBP to some ENSO index? Please clarify.

Figure 1. a,b,d,e It would be good to include error bars in these panels as with panels c and f. Panels c, f would be better shown as a percentage change, or relative difference

Line 80: 'All the ESMs show positive values over the boreal fall, winter,' NBP is conventionally a positive number (with higher NBP indicating more productive ecosystems), so a positive sensitivity to ENSO would indicate more NBP during ENSO events. The authors state that a positive sensitivity of NBP to ENSO implies less carbon uptake during ENSO. Are you using a negative convention for NBP? The typically convention is that Net Ecosystem Exchange is negative (denoting removal of carbon from the atmosphere by ecosystems) whereas NBP is positive.

Line 84: ', in contrast to the ENSO magnitude peak in the boreal winter.' Is this peak shown somewhere?

Line 86: 'However, in a warm climate, this peak moves forward to January'. This peak is not very well constrained, given the variability in the models. So although, as visually presented, it appears to occur in January that dynamic seems largely driven by one anomalous model (BCC-CSM-1). I do not think your conclusion is well supported here.

Line 166:

It is not clear how we should interpret this results presented in Figure 2, given that the change in sensitivity is driven by different factors in different regions. It would be better to present first the results in figure 3, and then repeat the analysis in figure 2 but performed instead on the data presented in panels c and d of the current figure 3.

Line 122: 'Enhanced terrestrial carbon fluxes'. I presume you mean reduced NBP?

Line 132: Some of the model spread in figure 2 could be due to differences in modeled ENSO, not differences in the land surface response.

Line 179: This figure could be added to the SI.

Line 172-205: Much of this text is discussion, not results.

Line 222: The paper cited here (28) is quite poor and provides no evidence to support the statement made by the authors.

Line 222-227: These sentences are alarmist and unsupported. You are using multiple Earth-system models that predict these scenarios are unlikely to occur.

Line 239: control -> influence

Line 430: kg C or kg CO₂?

Reviewer #3 (Remarks to the Author):

The authors used CMIP5 model ensemble to indicate that the sensitivity of terrestrial carbon cycling to ENSO would increase in the future. A fundamental problem is that the authors' main conclusion would be seriously jeopardized by a large model spread in future simulations. Moreover, I have also challenged the meaningfulness of exploring carbon cycling sensitivities to ENSO. ENSO dynamics is a mixture of internal variability and changes from human interventions and natural forcing such as solar and volcanic eruptions. The impacts of ENSO on carbon cycling are mainly manifested in ENSO-induced climate change. Instead of directly considering the future climate change on carbon cycling, studying the temporal change in the ENSO-carbon cycle linkage would be further hindered by the wide recognition that the earth system models are often notorious for bad performance in reproducing the magnitude and phase of ENSO. I am not convinced that the MS has a wider implication and improved recognition for current climate-carbon cycle studies.

As shown in Figure 1, there is a large model spread in the magnitude of the ENSO sensitivity and its intra-seasonal variation particularly for the future RCP4.5 scenario. This leads me to challenge the use of future model projections in inferring the temporal change in carbon cycle sensitivity to ENSO. Most notably, the authors stated that the timing of delayed peak in the ENSO-related carbon flux would move forward from March in the historical period to January in the future model projection, however which seems to be mainly due to an anomalous high sensitivity value in January from the model BCC-CSM1-1.

When it came to the regional attribution, the Amazon and Australia were identified to be the two main regions contributing to enhanced GPP response to ENSO. This was consistent with previous studies focusing on the historical period such as Wang et al. (2013) and Ahlström et al. (2015) showing that these regions significantly contribute to global carbon cycling in terms of trend and variability. But it should not be used as evidence to demonstrate the robustness of changes in the GPP response to ENSO in the tropics leading to global changes in the GPP response to ENSO. As also stated above, the current regional attribution was based on an ensemble of model simulations with a large (unknown) model spread in the simulation of intra-seasonal variability of carbon fluxes sensitivity to ENSO.

Lastly, the authors developed a decomposition framework to show that both GPP sensitivity to temperature and temperature sensitivity to ENSO made substantial contribution to enhanced GPP sensitivity to ENSO. The frustrating thing about the conclusions is that the analysis did not go much further than the decomposition and did not offer any insights into the underlying mechanisms. I would believe that the enhanced sensitivity of terrestrial production to temperature is not just simply attributed to the higher vapor-pressure deficit with temperature, as mentioned by the authors. This also applies to the issue of enhanced temperature responses to ENSO in a warmer climate.

References

Wang, W. L. et al. Variations in atmospheric CO₂ growth rates coupled with tropical temperature. *P. Natl. Acad. Sci. U.S.A.* 110, 15163–15163 (2013).
Ahlström, A. et al. The dominant role of semi-arid ecosystems in the trend and variability of the land CO₂ sink. *Science* 348, 895–899 (2015).

Reviewer #4 (Remarks to the Author):

This is a multi-model analysis, designed to determine how the response of the global carbon cycle to the El Niño/Southern Oscillation will change as a result of global warming. The authors employ nine of the models from the Coupled Model Intercomparison Project phase 5 (CMIP5) in the pre-industrial and Extended Concentration Pathway (ECP4.5) configurations, and analyze the response of various carbon cycle parameters to ENSO. They find a significant enhancement in the ENSO-driven response in the future, driven primarily by changes in the Amazon in response to both enhanced productivity sensitivity to local temperature and to enhanced local temperature response to ENSO. They conclude that future climate variability may have more dramatic impacts on agricultural production than previously anticipated.

Overall comments:

I enjoyed this paper - it addresses a question of crucial significance to the field in a relatively novel way. I would like to see it published in *Nature Communications*, but believe the authors should do some cleaning up/condensing of the text and clarifying their analysis choices first. In particular, I was quite curious: if you run your analysis on the 2006-2100 period, how do the results compare? It makes sense to use the 2100-2300 stabilized period but this is not generally the most relevant timescale for impacts purposes, so I think it would be good to include a discussion of the 21st century as well.

Also the manuscript would probably benefit from proofreading by a native English speaker.

Specific comments:

Lines 34-35: Add a mention of the methods used to choose the models for analysis here.

Line 46: Really, productivity decreases over the entire tropics during El Niño? Why is there not a dipolar structure in productivity, as we see for temperature and precipitation?

Line 51: I don't think you need to mention the fact that CO₂ has a greenhouse effect here!

Lines 54-55: The first two sentences in this paragraph are redundant, suggest consolidating into one.

Line 56: A reference to the Collins et al. (2010) *Nature Geoscience* review on ENSO and greenhouse warming would be useful here.

Lines 62-63: Sentence beginning "Depending on varying climate states..." needs a reference.

Lines 63-64: Sentence beginning "In summary, ..." isn't necessary and can be deleted.

Line 69: You need to specify why the ECP4.5 and not RCP4.5 is used here; there is a nice explanation in the Methods section, suggest moving that text up to here.

Line 72: How many CMIP5 models were run with active carbon cycles? The Methods says there are 9 with both carbon cycles and 2100-2300 extensions, but I wasn't sure if there were more BGC-

enabled models which were only run to 2100.

Line 77: Define "net biome production" here, and how it generally relates to both overall carbon fluxes into the atmosphere and the state of ENSO.

Line 78: I found the choice of global regressions in Figure 1 confusing, since all of the motivating references earlier on relate to the tropics, and given the tropical/subtropical focus of the regional analyses later on. Suggest regenerating this figure using only the 40N-40S window used for other analyses.

Line 79: No observations are actually plotted in Figure 1, so this statement is not supported on the basis of the analysis here.

Line 98: Suggest deleting mention of fires, since their influence is not included in Fig. S1.

Lines 107-8: Why do ESMs underestimate fire/heterotrophic respiration influences?

Lines 111-115: I see what the authors are trying to say here, but the phrasing is quite confusing. Please rephrase these sentences to more clearly explain the main point (that even models with insignificant mean GPP changes sometimes pass the threshold for significance during boreal winter).

Line 124: This is incorrect. The analyses in Figure 2 are regression coefficients of GPP against ENSO for individual regions, and their values cannot simply be added together to give the overall global sensitivity. Please rephrase to indicate this fact.

Lines 139-143: Again this can be condensed for clarity.

Line 148: "similar" structure is not that similar in North America or in southern South America, what is going on in those locations?

Paragraphs beginning at lines 153 and 159: can be combined

Line 166: This was confusing. The remote impact of ENSO *is* altered under greenhouse warming, as the authors show in the very next paragraph. Do you mean that even *if* the remote impact were not to change, the GPP sensitivity would still enhance ENSO/carbon cycle feedbacks? If so please say that instead.

Lines 198-9: This is an interesting result! I'd just suggest including references for the negative evaporation/soil moisture feedbacks in soil moisture-limited regimes.

Lines 258-9: Other appropriate references to include on internal ENSO variability and climate change: Wittenberg 2009 GRL, Stevenson et al. 2010, 2012 J. Climate.

Line 285: Specify explicitly that the partial regression coefficients here are derived from a multiple linear regression of GPP on T, P

Figure 1:

- Font sizes don't match between panels a/d, b/c/e/f
- Are the regressions plotted here simultaneous in time?
- How many ensemble members are used for each model?
- Panel g: PI, ECP should be plotted in different colors or intensities, right now they're hard to distinguish. Also there should be error bars on the ECPs as well

Figure 2:

- Label inset boxes
- X tick labels for regions need to be easier to understand: suggest adding some kind of legend for decoding the abbreviations

Figure 3:

- Suggest eliminating X tick labels on panels a-d, since they get mixed up with the superscripts on the gamma's
- Why are the regression coefficients missing over N. Africa?

Figure 4: Switch the sign of the color bar in panels c,d - right now it is confusing.

Supp Figure 1:

- Add panel labels (a,b,c)
- The legend is hard to read; make bigger and/or bolder text
- The NBP and GPP colors are nearly indistinguishable, at least when I printed the figure. Make one dashed and/or a different color

Supp Figures 2, 4: Same comments as Figure 2

Reviewer #1 (Remarks to the Author):

This is an important and interesting subject, and despite my generally critical comments to follow, I think this is excellent work and worthy of publication after some additional work. This topic has not been addressed in a significant way since Wang and Schimel's 2003 review, but as evidence grows for the powerful effect of El Nino on tropical carbon cycling, it is of growing concern relative to the future.

My concern is as follows. The authors do a thorough analysis of the CMIP ensemble but don't address either its significant divergence, suggesting fundamental disagreement over mechanism nor comparison with observations.

Response: We appreciate the Reviewer #1 for encouraging comments. The reviewer's comments were fully incorporated in the revised manuscript. Our responses to the specific comments are as follows:

This raises two issues that have to be dealt with. First, in contrast to physical ensembles, there is little evidence that the central tendency of the carbon ensemble is more skillful than the individual models. In fact, model evaluations show the ensemble to be significantly biased relative to observational constraints (papers by Hoffman, Cox and Friedlingstein). While these results have challenges of their own, they suggest that use of the ensemble is not as well justified as for physical climate.

Response: We have added analysis using individual model results into Supplementary Figures (Fig. A1 and A2). As a result, most of the models show significantly enhanced ENSO-carbon coupling in the future projection (Fig. 1c and others), as well as the ensemble mean.

In addition, it is clear that most of the ESMs simulate enhanced surface temperature response to ENSO in the future projection, especially in Amazonia (Supplementary Fig. 4 and 6). Only one ESM shows positive GPP anomalies related to enhanced in temperature response to ENSO, but other eight models show negative GPP anomalies. Therefore, we suggest here that there is a clear tendency to enhanced GPP anomalies related to ENSO under a warm climate in spite of the large model spread.

In addition, we have analyzed the ESMs using "Emergent constraint" approach (Cox et al. 2013) (Fig. A3). It is clear that γ_{NBP}^{ENSO} in the ESMs have a strong linear relationship (Correlation=0.86, $P<0.01$) between pre-industrial and ECP4.5 scenario. By using CO_2 growth rate observed in Mauna Loa at Hawaii in recent 46 years, sensitivity to ENSO ($\gamma_{CO_2}^{ENSO}$) is -1.2 ± 0.4 PgC yr⁻¹, while MME for pre-industrial is -1.35 ± 0.25 PgC yr⁻¹. It is clear that both MME and corrected one by the emergent constraint show the intensified ENSO-related carbon flux under greenhouse warming. In particular, all models, simulating the observed range of ENSO-related carbon flux, show the intensified coupling with ENSO.

Second, the authors focus in on GPP as the primary driver of response to El Nino. This is also not justified relative to observations, and calls the exercise into question. The literature demonstrates conclusively that a large part of the El Nino cycle is due to variation in fire, and this both affects NEE directly, and should influence it in a lagged fashion, as burned areas regrow. Omitting this is not justified (papers by Francey, van

der Werf, Bloom and many others). Second, some studies (papers by Anderegg and Clark) suggest a prominent role for respiration in tropical El Nino responses. Taken together, this conflicts with the model's primary response occurring through GPP and suggests the models, even when they replicate observed patterns are doing so for the wrong reasons. Forthcoming papers from the OCO-2 science team, as presented at AGU last year, will show even greater complexity (poster presentations by Liu, Chatterjee and others).

Response: As the Reviewer #1 pointed out, not only GPP but also fire and respiration have a role in the interannual variability of NEE (NBP) anomaly. We have analyzed carbon fluxes due to respiration and nature fire with respect to ENSO (Supplementary Fig. 2). As a result, there are significant changes in fire emission related to ENSO between pre-industrial and ECP4.5 scenario (Supplementary Fig. 2d), but only four models release simulation results of nature fire carbon emission. In addition, both of carbon fluxes due to R_a (autotrophic respiration) and R_h (heterotrophic respiration) do not show significant changes (Supplementary Fig. 2b,d), at least in the CMIP5 models. Unfortunately, the CMIP5 models tend to underestimate the effects of the natural fire and the respiration. Therefore, it may have a limitation to estimate the changes in wild fire and respiration under greenhouse warming, as the reviewer pointed out. However, qualitatively, the effects of the wild fire and respiration associated with ENSO can be enhanced under greenhouse warming due to the dryer and warmer conditions over the tropical lands, which can further enhance the intensification of the ENSO-related terrestrial carbon flux that we are emphasizing in this paper. We have added more discussion related to current state of simulating nature fire and respiration in the ESMs as follows:

(P10 L232-242) "In contrast to GPP, R_a and R_h do not show significant changes between two scenarios (Supplementary Fig. 2). Indeed, there are various parameterizations for R_h related to temperature variation in the CMIP5 ESMs³⁴ and it is reported that the current land surface models have a systematic problem to capture temperature sensitivity of respiration, especially linked to long-term temperature variability³⁵. In order to estimate future changes in R_h response to ENSO, the model parameterization for R_h should be further validated and investigated³⁶. In addition, it has been reported that current ESMs, including process-based fire scheme, tend to underestimate the effects of wild fires^{18,37,38}. These imply our results can be overlooked contributions of changes in R_h and wild fire changes related to $\Delta\gamma_{Temp}^{ENSO}$, and it might further enhance the intensification of the ENSO-related terrestrial carbon flux, rather than what we reported here."

As a result of these two concerns, in contrast to current diagnostic papers on the physical system, where a certain measure of skill can be taken for granted and is established in the literature, the carbon cycle models cannot be assumed to have similar levels of skill, and this analysis itself suggests they may all respond to El Nino-type variability incorrectly.

A diagnostic paper on the response of the CMIP ensemble to El Nino variability is of great interest and value, but drawing scientific conclusions without dealing with the potentially near-total lack of correct mechanism is unwarranted. It may well be, indeed,

I expect that the authors conclusions about carbon-water coupling remain robust to these model flaws, but the paper is misleading if not presented in the perspective of how immature and erroneous these models actually are.

Response: In newly added Supplementary Fig. 10, surface heat budget does not show clear changes in future projection, while surface temperature anomalies related to ENSO (γ_{Temp}^{ENSO}) is enhanced over most of the tropics (Supplementary Fig. 9). On the other hand, previous studies found that there is a high possibility of stronger heat waves over tropics when soil moisture depletion would occur in future projection (Seneviratne et al. 2010, 2013). When land has enough soil moisture, land and atmosphere exchange their heats through latent heat flux. However, if in the soil-moisture-limited regime, the sensible heat flux is a dominant component for exchanging heat between land and atmosphere, thereby near land surface can easily warmer by heat waves under El Nino-induced forcing.

Indeed, it is being considered that soil moisture likely reduced under greenhouse warming, especially in Amazonia (Fig. 4b). In this sense, we have analyzed soil moisture depletion in Amazonia and surface temperature anomalies related to ENSO (γ_{Temp}^{ENSO}) in several scenarios of CMIP5 (Fig. A4). Most of the ESMs, except for MIROC-ESM and IPSL-CM5A-MR, which do not show significant result at 95% confidence level in Fig. 1c, well represent a negative tendency in soil moisture and enhanced surface temperature anomalies related to ENSO (positive $\Delta\gamma_{Temp}^{ENSO}$) under further greenhouse warming. By this enhanced land-atmosphere interaction, stronger heat waves may also cause terrestrial productivity variation much more as we present in this study. As the reviewer pointed out, there is a large uncertainty in the carbon-water coupling of the current models. However, this result suggests common tendencies for soil moisture depletion, and the enhanced temperature responses to ENSO under greenhouse warming, and they are physically linked.

References

- Cox, P. M. *et al.* Sensitivity of tropical carbon to climate change constrained by carbon dioxide variability. *Nature* **494**, 341–344 (2013).
- Seneviratne, S. I. *et al.* Investigating soil moisture-climate interactions in a changing climate: A review. *Earth-Sci. Rev.* **99**, 125–161 (2010).
- Seneviratne, S. I. *et al.* Impact of soil moisture-climate feedbacks on CMIP5 projections: First results from the GLACE-CMIP5 experiment. *Geophys. Res. Lett.* **40**, 5212–5217 (2013).

Figure A1 | NBP sensitivities to ENSO in individual models. Regression coefficients of carbon flux anomalies in 40°S–40°N on the DJF Niño3.4 index for NBP based on the pre-industrial experiment (blue) and ECP4.5 (red). Shaded area indicates 95% confidence levels calculated using bootstrap method.

Figure A2 | GPP sensitivities to ENSO in individual models. Regression coefficients of carbon flux anomalies in 40°S–40°N on the DJF Niño3.4 index for GPP based on the pre-industrial experiment (blue) and ECP4.5 (red). Shaded area indicates 95% confidence levels calculated using bootstrap method.

Figure A3 | Emergent constraint on ENSO-related carbon flux. Gray and black dots show the individual ESMs and MME for γ_{NBP}^{ENSO} in pre-industrial (x-axis) and ECP4.5 (y-axis) scenarios. Light red shading indicates $\gamma_{CO_2 \text{ growth rate}}^{ENSO}$ by using Mauna Loa CO₂ observation at 95% confidence level calculated using bootstrap method. Dashed line means a linear relationship between two experiments in CMIP5 ($P < 0.01$) and thick lines are 95% confidence level calculated using bootstrap method.

Figure A4 | Soil moisture depletion and surface temperature anomalies related to ENSO in Amazonia. SONDJF mean soil moisture depletion in Amazonia (100° – 165° W, 20° S– 10° N) for pre-industrial (blue), historical (green), RCP4.5 (orange) and ECP4.5 (red) and sensitivity of SONDJF mean surface temperature in Amazonia to ENSO ($\gamma_{\text{Temp}}^{\text{ENSO}}$) based on the linear regression against December–February (DJF) Niño3.4 index. Lines indicates 95% confidence levels calculated using bootstrap method.

Reviewer #2 (Remarks to the Author):

Kim et al. report the results of an assessment of the sensitivity of the carbon cycle to the El Niño Southern Oscillation using output from the CMIP5 models, both under current climate conditions and future climate change. They show a consistent response of the CMIP5 models in terms of the sensitivity to ENSO events, and the seasonal distribution. More importantly, they show an increase in the sensitivity under future climate change. The paper is well written, the analysis and methodology well described, and the results will no doubt be of interest to the readers of Nature Communications.

Response: We thank the Reviewer for his/her encouraging and constructive comments. The Reviewer's comments were very helpful for significantly improving the manuscript.

General comments:

It is not clear how the sensitivity metric is defined for NBP. For ENSO-Temp relationships it is defined based on anomalies, but the definition for NBP is never given, other than saying 'we estimated the sensitivity of net biome production (NBP) to ENSO on the basis of the linear regression in the ESMs.' Are NBP anomalies used, and normalized units? Could the sensitivity increase just because NBP increases?

Response: As same as temperature, interannual NBP anomalies are used for sensitivity analysis. Moreover, we did not use any normalization for carbon fluxes such as NBP. In order to clarify this issue, we have edited that sentence as follows:

(P4 L82-85) "we estimated the sensitivity of net carbon flux from land (Net Biome Production; NBP) in 40°S–40°N to ENSO ($\gamma_{\text{NBP}}^{\text{ENSO}}$; $\text{PgC yr}^{-1} \text{ } ^\circ\text{C}^{-1}$) based on the linear regression of NBP anomalies against December–February (DJF) Niño3.4 index."

Why did the authors use RCP4.5? Presumably RCP8.5 would give clearer results? It would be informative to compare results from the two scenarios.

Response: We agree that RCP8.5 would give more certain result in terms of greenhouse warming effect on ENSO-related carbon cycle; however, only three models have the stabilized period for RCP8.5, so it is difficult to show a credible model consistency from a small sample.

Detailed comments:

The sentences from Line 54 to 61 repeat the same statement in different ways.

Response: We have revised that sentences as follows:

(P3 L53-58) "The response of the terrestrial carbon cycle to ENSO could be changed in a warm climate. For example, it has been reported that ENSO impacts would change under greenhouse warming^{9–13} because changes in the background mean states can modulate ENSO responses such as atmospheric circulation patterns, surface temperature, and precipitation. These changes in these atmospheric conditions associated with ENSO could also affect their impacts on the terrestrial ecosystem and carbon cycle."

Line 61: It is unclear why mean climate states would change the sensitivity. Are the authors suggesting that the response is a non-linear function of mean climate state?

Response: Because terrestrial production sensitivity to temperature tends to follow the parabolic curve and having an optimum range at the center (Yamori et al., 2014), the terrestrial response to anomalous temperature largely depends on the mean temperature.

Therefore, the mean climate change leads to changes in sensitivity of terrestrial carbon flux to local temperature anomaly.

We have revised that sentence as follows:

(P3 L58-61) “In addition, mean climate changes lead to changes in the sensitivity of the terrestrial carbon flux to local temperature anomaly¹⁴⁻¹⁶, thereby changes in sensitivity depending on varying climate states would directly connected to the strength of the terrestrial response to ENSO.”

Line 72: ‘as well as’ 200 years of ECP4.5

Response: corrected.

Line 78: ‘linear regression in the ESMs’. It is not clear what you mean here. I presume the linear regression of the relationship between NBP to some ENSO index? Please clarify.

Response: We used DJF Niño3.4 index as described in Method section. We have clarified it in the revised manuscript as follows.

(P4 L82-85) “we estimated the sensitivity of net carbon flux from land (Net Biome Production; NBP) in 40°S–40°N to ENSO ($\gamma_{\text{NBP}}^{\text{ENSO}}$; $\text{PgC yr}^{-1} \text{ } ^\circ\text{C}^{-1}$) based on the linear regression of NBP anomalies against December–February (DJF) Niño3.4 index.”

Figure 1. a,b,d,e It would be good to include error bars in these panels as with panels c and f.

Panels c, f would be better shown as a percentage change, or relative difference

Response: The percentage change is sometimes not good because carbon flux anomalies in pre-industrial simulation show almost zero in ENSO non-active season such as JJA. However, we have added a description about percentage change in ENSO active season in the revised manuscript as follows:

(P5 L94-97) “This intensification ($\Delta\gamma_{\text{NBP}}^{\text{ENSO}}$) is clearly found during September to the following February (SONDJF; Fig. 1b), relatively 44.9% for SON and 44.1% for DJF, when ENSO anomalies are strong.”

Line 80: ‘All the ESMs show positive values over the boreal fall, winter,’

NBP is conventionally a positive number (with higher NBP indicating more productive ecosystems), so a positive sensitivity to ENSO would indicate more NBP during ENSO events. The authors state that a positive sensitivity of NBP to ENSO implies less carbon uptake during ENSO. Are you using a negative convention for NBP? The typically convention is that Net Ecosystem Exchange is negative (denoting removal of carbon from the atmosphere by ecosystems) whereas NBP is positive.

Response: Yes, we used a negative convention for NBP, GPP, R_a , and R_h . In the revised manuscript, we changed them to the positive convention for NBP to avoid confusion.

Line 84: ‘, in contrast to the ENSO magnitude peak in the boreal winter.’ Is this peak shown somewhere?

Response: We did not show the seasonal evolution of ENSO magnitude, but it is well-known that ENSO phenomena have their maximum in the boreal winter. In order to clarify, we have revised that sentence with some references as follows:

(P4 L89-90) “It is also found that the maximum NBP response to ENSO appears in the following March whereas the ENSO magnitude is usually at the peak in the boreal winter^{4,16,17}”

Line 86: ‘However, in a warm climate, this peak moves forward to January’. This peak is not very well constrained, given the variability in the models. So although, as visually presented, it appears to occur in January that dynamic seems largely driven by one anomalous model (BCC-CSM-1). I do not think your conclusion is well supported here.
Response: We agree with your comment. We have deleted that sentence.

Line 166:

It is not clear how we should interpret this results presented in Figure 2, given that the change in sensitivity is driven by different factors in different regions. It would be better to present first the results in figure 3, and then repeat the analysis in figure 2 but performed instead on the data presented in panels c and d of the current figure 3.

Response: We appreciate the reviewer’s good suggestion. We have changed the order of the figures in the manuscript as following your comment. In the revised manuscript, the previous Figure 2 is moved to Figure 3. Also, we have added a supplementary figure which includes $\Delta Y_{GPP}^{Temp} Y_{Temp}^{ENSO}$ and $\Delta Y_{Temp}^{ENSO} Y_{GPP}^{Temp}$ in each region (Supplementary Fig. 8; Fig. B1).

Line 122: ‘Enhanced terrestrial carbon fluxes’. I presume you mean reduced NBP?

Response: We have edited that sentence as follows:

(P8 L167-170) “Of these regions, Amazonia, Australia, and Insular Southeast Asia considerably contribute to the reduced carbon fluxes out of atmosphere due to GPP on land, i.e., -0.31 , -0.18 , and -0.13 PgC yr⁻¹ °C⁻¹, respectively (Fig. 3).”

Line 132: Some of the model spread in figure 2 could be due to differences in modeled ENSO, not differences in the land surface response.

Response: Since we used the linear regression, the difference of ENSO magnitude among the models is already considered (anomalies per unit Niño3.4 SST (1K)), but the pattern differences of the ENSO and its teleconnection are not considered. We have revised that sentence including a role of uncertainty from ENSO teleconnection simulations.

Line 179: This figure could be added to the SI.

Response: We added the figure in Supplementary Fig. 10 (Fig. B2).

Line 172-205: Much of this text is discussion, not results.

Response: Because the main figure, such as Fig. 4, should be included in main text based on the guide of this journal, we had to describe relevant contents in Result section, not in Discussion section.

In addition, we have changed sub-headings that break up the main text as follows:

(P8 L184-185) “**Possible mechanism for the enhanced temperature responses to ENSO in a warmer climate**”

Line 222: The paper cited here (28) is quite poor and provides no evidence to support the statement made by the authors.

Response: We have cited more references which deal with tree mortality risk due to drought such as Anderegg et al., 2013; Anderegg et al., 2016; Millar and Stephenson 2015.

Line 222-227: These sentences are alarmist and unsupported. You are using multiple Earth-system models that predict these scenarios are unlikely to occur.

Response: We agree that RCP and ECP scenarios are idealized experiments which are unlikely to occur in future. However, we would discuss a possible role of extreme events on the long-term carbon flux.

Line 239: control -> influence

Response: corrected.

Line 430: kg C or kg CO₂?

Response: kgC. We have edited as kgC.

References

Yamori, W., K., Hikosaka, D. A., Way, Temperature response of photosynthesis in C₃, C₄, and CAM plants: temperature acclimation and temperature adaptation. *Photosynth. Res.* **119**, 101–117 (2014).

Booth, B. B. B. et al. High sensitivity of future global warming to land carbon cycle processes. *Environ. Res. Lett.* **7**, 1–8 (2012).

Wang, X. H. et al. A two-fold increase of carbon cycle sensitivity to tropical temperature variations. *Nature* **506**, 212–215 (2014).

Liu, Y. W. et al. Changes in interannual climate sensitivities of terrestrial carbon fluxes during the 21st century predicted by CMIP5 Earth System Models. *J. Geophys. Res.* **121**, 903–918 (2016).

Anderegg, W. R. L., Kane, J. M. & Anderegg, L. D. L. Consequences of widespread tree mortality triggered by drought and temperature stress. *Nat. Clim. Change* **3**, 30–36 (2013).

Anderegg, W. R. L., Klein, T., Bartlett, M., Sack, L., Pellegrini, A. F. A., Choat, B. & Jansen, S. Meta-analysis reveals that hydraulic traits explain cross-species patterns of drought-induced tree mortality across the globe. *Proc. Natl Acad. Sci. USA* **113**, 5024–5029 (2016).

Millar, C. I. & Stephenson, N. L. Temperate forest health in an era of emerging megadisturbance. *Science* **349**, 823–826 (2015).

Figure B1 | Regional contribution in Fig 2c,e. Regional contribution of the changes the surface temperature response to ENSO ($\Delta y_{Temp}^{ENSO} \gamma_{GPP}^{Temp}$; left bars) and GPP sensitivity to temperature ($\Delta y_{GPP}^{Temp} \gamma_{Temp}^{ENSO}$; right bars). Each bar shows the MME results based on each region, as marked on the map. Error bars indicate 95% confidence levels based on bootstrap estimation.

Figure B2 | γ_{Temp}^{ENSO} and total summation of radiation in 40°S–40°N. (a) MME γ_{Temp}^{ENSO} for pre-industrial (blue) and ECP4.5 (red) scenario. (b) total summation of radiation that downward/upward long/short wave radiation and sensible/latent heat in the pre-industrial experiment and ECP4.5 ($W m^{-2} \text{ } ^\circ C^{-1}$).

Reviewer #3 (Remarks to the Author):

The authors used CMIP5 model ensemble to indicate that the sensitivity of terrestrial carbon cycling to ENSO would increase in the future. A fundamental problem is that the authors' main conclusion would be seriously jeopardized by a large model spread in future simulations. Moreover, I have also challenged the meaningful of exploring carbon cycling sensitivities to ENSO. ENSO dynamics is a mixture of internal variability and changes from human interventions and natural forcing such as solar and volcanic eruptions. The impacts of ENSO on carbon cycling are mainly manifested in ENSO-induced climate change. Instead of directly considering the future climate change on carbon cycling, studying the temporal change in the ENSO-carbon cycle linkage would be further hindered by the wide recognition that the earth system models are often notorious for bad performance in reproducing the magnitude and phase of ENSO. I am not convinced that the MS has a wider implication and improved recognition for current climate-carbon cycle studies.

Response: We thank the Reviewer #3 for his/her time and critical comments. Our responses to the specific comments are as follows:

1. Model spread

In the revised manuscript, we tried to support our arguments not only from multi-model ensemble (MME) but also from individual models. We would demonstrate the points that the most CMIP5 models have common tendency under the large spread of the model simulation. For example, eight of nine ESMs tend to simulate enhanced NBP response to ENSO in SONDJF (Fig. 1c). Even though there are model spread in terms of magnitude in future changes, tendency of models is quite robust in order to suggest future enhancement of ENSO-related carbon cycle.

In addition, in order to evaluate the spread in ESMs, we have analyzed the ESMs using "Emergent constraint" approach (Cox et al. 2013) (Fig. C1). It is clear that $\gamma_{\text{NBP}}^{\text{ENSO}}$ in the ESMs have a strong linear relationship (Correlation=0.86, $P<0.01$) between pre-industrial and ECP4.5 scenario, suggesting spread in the ESMs is at least limited to inter-model differences, not much diverse by climate state. By using CO_2 growth rate observed in Mauna Loa at Hawaii in recent 46 years, sensitivity to ENSO ($\gamma_{\text{CO}_2}^{\text{ENSO}}$) is $-1.2\pm 0.4 \text{ PgC yr}^{-1}$, while MME for pre-industrial is $-1.35\pm 0.25 \text{ PgC yr}^{-1}$. It is clear that both MME and corrected one by the emergent constrain show the intensified ENSO-related carbon flux under greenhouse warming. In particular, all models, simulating observed range of ENSO-related carbon flux, show the intensified coupling with ENSO.

2. ENSO performance in the CMIP5 ESMs

We agree with the reviewer 3 on the limitation of the modeling studies using incomplete climate models. As compared to observation, the CMIP ESMs have some biases in terms of simulating ENSO. In spite of the systematic biases, many studies agree that the CMIP5 climate models are reasonably capturing main characteristics of ENSO (e.g. Kug et al. 2012) and its tropical impacts (Weare 2013). Importantly, we have understood what are systematic biases in ENSO simulation and what cause the systematic biases in the CMIP5 models (e.g. Ham and Kug 2015). In addition, the

previous study showed that the CMIP5 ESMs well represent ENSO-related carbon cycle in terms of phase, magnitude and peak timing (Kim et al. 2016). So, we believe that in spite of some model biases the current results can give important implication on ENSO-carbon cycle coupling and future changes in the global cycle because the state-of-the-art climate models simulate common tendency.

As shown in Figure 1, there is a large model spread in the magnitude of the ENSO sensitivity and its intra-seasonal variation particularly for the future RCP4.5 scenario. This leads me to challenges the use of future model projections in inferring the temporal change in carbon cycle sensitivity to ENSO. Most notably, the authors stated that the timing of delayed peak in the ENSO-related carbon flux would moves forward from March in the historical period to January in the future model projection, however which seems to mainly due to an anomalous high sensitivity value in January from the model BCC-CSM1-1.

Response: In order to check the larger one model contribution (e.g. BCC-CSM1-1), MME result and its 95% confidence levels are recalculated using the eight models, except for BCC-CSM1-1 (Fig. C2). Even though MME values are slightly changed, but the main result does not change without BCC-CSM1-1, indicating the robustness of the results. We agree with the Reviewer's comment on the delayed peak, so we deleted the discussion on that.

When it came to the regional attribution, the Amazon and Australia were identified to be the two main regions contributing to enhanced GPP response to ENSO. This was consistent with previous studies focusing on the historical period such as Wang et al. (2013) and Ahlström et al. (2015) showing that these regions significantly contribute to global carbon cycling in terms of trend and variability. But it should not be used as evidence to demonstrate the robustness of changes in the GPP response to ENSO in the tropics leading to global changes in the GPP response to ENSO. As also stated above, the current regional attribution was based on an ensemble of model simulations with a large (unknown) model spread in the simulation of intra-seasonal variability of carbon fluxes sensitivity to ENSO.

Response: In newly added Supplementary Fig. 4, 5 and 6, ΔY_{GPP}^{ENSO} , ΔY_{GPP}^{Temp} , γ_{Temp}^{ENSO} , and $\Delta \gamma_{Temp}^{ENSO} \gamma_{GPP}^{Temp}$ have significant values in Amazonia and Australia at 95% confidence level calculated using bootstrap method (Fig. C3, C4 and C5). Even though model spread, most of the models tend to simulate enhanced ENSO-related carbon cycle at least in Amazonia and Australia.

Lastly, the authors developed a decomposition framework to show that both GPP sensitivity to temperature and temperature sensitivity to ENSO made substantial contribution to enhanced GPP sensitivity to ENSO. The frustrating thing about the conclusions is that the analysis did not go much further than the decomposition and did not offer any insights into the underlying mechanisms. I would believe that the enhanced sensitivity of terrestrial production to temperature is not just simply attributed to the higher vapor-pressure deficit with temperature, as mentioned by the authors. This also applies to the issue of enhanced temperature responses to ENSO in a warmer climate.

Response: In the revised manuscript, we tried to discuss the underlying mechanism more, and provide more supporting materials.

In the current land surface models, a relationship between GPP and temperature just follows simplified function such as vapor-pressure deficit (VPD), not including others such as photo-degradation and changes in insolation. Moreover, Shao et al. (2016) argued that VPD mostly explains interannual variability of NEE, based on 65 eddy covariance measurement sites, among photosynthetically active radiation, air temperature, soil temperature, water balance index, maximum photosynthetic rate, apparent quantum yield, water use efficiency, reference respiratory rate at 10°C and temperature sensitivity.

References

Wang, W. L. et al. Variations in atmospheric CO₂ growth rates coupled with tropical temperature. *P. Natl. Acad. Sci. U.S.A.* 110, 15163–15163 (2013).

Ahlström, A. et al. The dominant role of semi-arid ecosystems in the trend and variability of the land CO₂ sink. *Science* 348, 895–899 (2015).

References

Kug, J.-S., Ham, Y.-G., Lee, J.-Y. & Jin, F.-F. Improved simulation of two types of El Niño in CMIP5 models. *Environ. Res. Lett.* 7, 034002 (2012).

Weare, B. C. El Niño teleconnections in CMIP5 models. *Clim. Dyn.* 41, 2165–2177 (2013).

Ham, Y.-G. & Kug, J.-S. Improvement of ENSO simulation based on intermodel diversity. *J. Clim.* 28, 998-1015 (2015).

Kim, J.-S., Kug, J.-S., Yoon, J.-H. & Jeong, S.-J. Increased atmospheric CO₂ growth rate during El Niño driven by reduced terrestrial productivity in the CMIP5 ESMs. *J. Clim.* 29, 8783-8805 (2016).

Shao, J., et al. Direct and indirect effects of climatic variations on the interannual variability in net ecosystem exchange across terrestrial ecosystems. *Tellus B* 68, 30575 (2016).

Figure C1 | Emergent constraint on ENSO-related carbon flux. Gray and black dots show the individual ESMs and MME for $\gamma_{\text{NBP}}^{\text{ENSO}}$ in pre-industrial (x-axis) and ECP4.5 (y-axis) scenarios. Light red shading indicates $\gamma_{\text{CO}_2 \text{ growth rate}}^{\text{ENSO}}$ by using Mauna Loa CO₂ observation at 95% confidence level calculated using bootstrap method. Dashed line means a linear relationship between two experiments in CMIP5 ($P < 0.01$) and thick lines are 95% confidence level calculated using bootstrap method.

Figure C2 | NBP sensitivities to ENSO without BCC-CSM1-1. Same with Figure 1, but regardless BCC-CSM1-1 (only using other eight models).

Figure C3 | Changes in GPP response to ENSO ($\Delta\gamma_{GPP}^{ENSO}$). (a–i) Differences in the regression coefficient of SONDJF GPP anomalies on the DJF Niño3.4 index between the pre-industrial experiment and ECP4.5 for individual models. (j) MME result for changes in GPP response to ENSO. Gray area indicates non-significant region at 95% confidence level calculated using bootstrap method.

Figure C4 | Contribution of the changes in the GPP sensitivity to temperature ($\Delta \gamma_{GPP}^{Temp} \times \gamma_{Temp}^{ENSO}$) (a-i) $\Delta \gamma_{GPP}^{Temp} \times \gamma_{Temp}^{ENSO}$ for individual models. (j) MME result for $\Delta \gamma_{GPP}^{Temp} \times \gamma_{Temp}^{ENSO}$. Gray area indicates non-significant region at 95% confidence level calculated using bootstrap method.

Figure C5 | Contribution of the changes in the temperature response to ENSO
 ($\Delta \gamma_{Temp}^{ENSO} \gamma_{GPP}^{Temp}$) (a-i) $\Delta \gamma_{Temp}^{ENSO} \gamma_{GPP}^{Temp}$ for individual models. (j) MME result for $\Delta \gamma_{Temp}^{ENSO} \gamma_{GPP}^{Temp}$. Gray area indicates non-significant region at 95% confidence level calculated using bootstrap method.

Reviewer #4 (Remarks to the Author):

This is a multi-model analysis, designed to determine how the response of the global carbon cycle to the El Niño/Southern Oscillation will change as a result of global warming. The authors employ nine of the models from the Coupled Model Intercomparison Project phase 5 (CMIP5) in the pre-industrial and Extended Concentration Pathway (ECP4.5) configurations, and analyze the response of various carbon cycle parameters to ENSO. They find a significant enhancement in the ENSO-driven response in the future, driven primarily by changes in the Amazon in response to both enhanced productivity sensitivity to local temperature and to enhanced local temperature response to ENSO. They conclude that future climate variability may have more dramatic impacts on agricultural production than previously anticipated.

Response: We appreciate the Reviewer #4 for encouraging and careful comments. The reviewer's comments were fully incorporated in the revised manuscript. Our responses to the specific comments are as follows:

Overall comments:

I enjoyed this paper - it addresses a question of crucial significance to the field in a relatively novel way. I would like to see it published in Nature Communications, but believe the authors should do some cleaning up/condensing of the text and clarifying their analysis choices first. In particular, I was quite curious: if you run your analysis on the 2006-2100 period, how do the results compare? It makes sense to use the 2100-2300 stabilized period but this is not generally the most relevant timescale for impacts purposes, so I think it would be good to include a discussion of the 21st century as well. Also the manuscript would probably benefit from proofreading by a native English speaker.

Response: As the reviewer suggested, we checked the RCP4.5 scenario, the 2006-2100 period. The result showed that the NBP anomalies related to ENSO are intensified under greenhouse warming, though some models show larger uncertainty, possibly due to strong time-varying trend (Fig. D1a,b). These results strongly support our argument in this paper. We added this figure in the supplementary.

Specific comments:

Lines 34-35: Add a mention of the methods used to choose the models for analysis here.

Response: We have revised abstract as follows:

(P2 L32-35) "We show here that the sensitivity of the terrestrial carbon flux to ENSO will be significantly enhanced under greenhouse warming (by approximately 50%) in comparison to the pre-industrial and future projections from the Earth System models providing Extended Concentration Pathway 4.5 scenario"

Line 46: Really, productivity decreases over the entire tropics during El Niño? Why is there not a dipolar structure in productivity, as we see for temperature and precipitation?

Response: As the Reviewer pointed out, the temperature and precipitation show a dipole pattern associated with ENSO in the tropics (Fig. D2). However, we only confine to the tropical land areas, they exhibit negative precipitation and positive temperature anomalies there. In particular, the temperature responses in the tropical land area are

quite uniform. Therefore, the terrestrial productivity anomalies can be the same sign to a large extent, rather than have a dipole pattern as shown in Fig. B.

On the other hand, for Africa, positive temperature and precipitation anomalies are simulated by the ESMs (Fig. D2). Therefore, terrestrial productivity anomalies have a complicated spatial pattern in Africa (Fig. D3).

Line 51: I don't think you need to mention the fact that CO₂ has a greenhouse effect here!

Response: Deleted.

Lines 54-55: The first two sentences in this paragraph are redundant, suggest consolidating into one.

Response: We have revised these sentences as follows:

(P3 L53-58) "The response of the terrestrial carbon cycle to ENSO could be changed in a warm climate. For example, it has been reported that ENSO impacts would change under greenhouse warming⁹⁻¹³ because changes in the background mean states can modulate ENSO responses such as atmospheric circulation patterns, surface temperature, and precipitation. These changes in these atmospheric conditions associated with ENSO could also affect their impacts on the terrestrial ecosystem and carbon cycle."

Line 56: A reference to the Collins et al. (2010) Nature Geoscience review on ENSO and greenhouse warming would be useful here.

Response: Thanks for good suggestion. We cited the reference.

Lines 62-63: Sentence beginning "Depending on varying climate states..." needs a reference.

Response: We have merged the sentence with former sentence which have relevant references as follows:

(P3 L58-61) "In addition, regional changes in mean climate lead to changes in the sensitivity of the terrestrial carbon flux to local temperature anomaly¹⁴⁻¹⁶, thereby changes in sensitivity depending on varying climate states would directly connected to the strength of the terrestrial response to ENSO."

Lines 63-64: Sentence beginning "In summary, ..." isn't necessary and can be deleted.

Response: Deleted.

Line 69: You need to specify why the ECP4.5 and not RCP4.5 is used here; there is a nice explanation in the Methods section, suggest moving that text up to here.

Response: We have moved the explanation for using ECP4.5 from method to manuscript as follows:

(P3 L65-78) "In this study, we examine future changes in the sensitivity of the terrestrial carbon cycle to ENSO by comparing pre-industrial and future projections (i.e., Extended Concentration Pathway 4.5: ECP4.5), is the extension of the Representative Concentration Pathways (RCP) 4.5 with continuous anthropogenic forcing until 2300, in the Earth System Models (ESMs), participated in the Coupled Model Intercomparison Project Phase 5 (CMIP5). It is difficult to estimate temporal changes of interannual sensitivities using Representative Concentration Pathways (RCP) scenario

because atmospheric conditions and carbon fluxes are strongly affected by anthropogenic forcing with considerable year-by-year variations and a time-varying trend with respect to ongoing climate change. To estimate more stable interannual sensitivities under the influence of anthropogenic climate change, we use a stabilized period of 200 years of ECP4.5 that is the idealized experiment for the 22nd and 23rd centuries; in this way, interannual sensitivities have statistical significance to estimate the anthropogenic effects on these sensitivities when comparing the sensitivities obtained using the pre-industrial experiment (see Methods).”

Line 72: How many CMIP5 models were run with active carbon cycles? The Methods says there are 9 with both carbon cycles and 2100-2300 extensions, but I wasn't sure if there were more BGC-enabled models which were only run to 2100.

Response: There are 25 individual models providing the RCP4.5 scenario. Among them, we used 9 models which provide ECP4.5 simulation. In addition, we chose just one model from several models having similar model structure. For example, IPSL-CM5A-LR and IPSL-CM5A-MR share the same model platform at different resolutions, so we only use IPSL-CM5A-MR. The GISS-E2-H-CC and GISS-E2-H have the similar dynamic core and physical parameterization, so we only use GISS-E2-H. We found that their results are quite similar.

Line 77: Define "net biome production" here, and how it generally relates to both overall carbon fluxes into the atmosphere and the state of ENSO.

Response: We have revised that sentence as follows:

(P4 L82-85) “we estimated the sensitivity of net carbon flux from land (Net Biome Production; NBP) in 40°S–40°N to ENSO ($\gamma_{\text{NBP}}^{\text{ENSO}}$; $\text{PgC yr}^{-1} \text{ } ^\circ\text{C}^{-1}$) based on the linear regression of NBP anomalies against December–February (DJF) Niño3.4 index.”

Line 78: I found the choice of global regressions in Figure 1 confusing, since all of the motivating references earlier on relate to the tropics, and given the tropical/subtropical focus of the regional analyses later on. Suggest regenerating this figure using only the 40N-40S window used for other analyses.

Response: As the reviewer suggested, we have revised Figure 1 but using only 40°N–40°S domain.

Line 79: No observations are actually plotted in Figure 1, so this statement is not supported on the basis of the analysis here.

Response: We have revised that sentence as follows:

(P4 L85-86) “This sensitivity is well simulated by CMIP5 ESMs and is consistent with previous modeling results in terms of phase and peak timing^{17,18}”

Line 98: Suggest deleting mention of fires, since their influence is not included in Fig. S1.

Response: We have added ENSO-related carbon flux due to wild fire in Supplementary Fig. 3.

Lines 107-8: Why do ESMs underestimate fire/heterotrophic respiration influences?

Response: Kim et al. 2016 argued that the CMIP5 ESMs tend to overestimate NPP response to ENSO and underestimate others, especially for wild fire (Kloster et al. 2012; Kloster and Lasslop, 2017). For fire influence, in Supplementary Fig. 3, carbon flux due to wild fire is $-0.17 \text{ PgC yr}^{-1}$ that only 12% as compared to carbon flux due to GPP anomalies ($-1.41 \text{ PgC yr}^{-1}$ in SONDJF). However, for example, 2.8 PgC was emitted by natural fires during the 1997/98 El Niño, whereas only 2.1 PgC yr^{-1} was emitted during the non-El Niño years of 2002–07 (Van der Werf et al. 2010). In addition, only four model (CESM1-CAM5, IPSL-CM5A-MR, MPI-ESM-LR, and NORESM1-M) simulates carbon flux due to wild fire. This indicates NBP simulations in other models do not include wild fire influences.

Lines 111-115: I see what the authors are trying to say here, but the phrasing is quite confusing. Please rephrase these sentences to more clearly explain the main point (that even models with insignificant mean GPP changes sometimes pass the threshold for significance during boreal winter).

Response: We have revised that sentence as follows:

(P5 L99-103) “The increase in the MME is also significant at the 95% confidence level, based on the bootstrap estimates. In addition to the MME, all individual models simulate stronger NBP sensitivities to ENSO under greenhouse warming as compared to those in the pre-industrial experiment (Fig. 1c). In particular, among nine ESMs, six (eight) models exhibit significantly stronger sensitivities of NBP at the 95% (90%) confidence level.”

Line 124: This is incorrect. The analyses in Figure 2 are regression coefficients of GPP against ENSO for individual regions, and their values cannot simply be added together to give the overall global sensitivity. Please rephrase to indicate this fact.

Response: We have changes the relative quantity to absolute amount of carbon flux as follows:

(P8 L167-170) “Of these regions, Amazonia, Australia, and equatorial Asia considerably contribute to the enhanced carbon fluxes from land to atmosphere, i.e., -0.31 , -0.18 , and $-0.13 \text{ PgC yr}^{-1}$, respectively (Fig. 3).”

Lines 139-143: Again this can be condensed for clarity.

Response: We have revised that paragraph as follows:

(P6 L127-130) “To understand how the ENSO-related carbon cycle is intensified under greenhouse warming, we decompose the GPP response to ENSO into the contributions of local temperature and precipitation because it is well-known that the terrestrial GPP is affected by local temperature and precipitation to a large extent (see Methods, Eq. 2)^{7,8,17,18}.”

Line 148: "similar" structure is not that similar in North America or in southern South America, what is going on in those locations?

Response: In these regions, $\gamma_{\text{GPP}}^{\text{Temp}}$ is opposite to general feature in the tropics (Fig. 3 in Kim et al., 2016). Therefore, the opposite response in these regions also means enhanced ENSO-related carbon flux, but in the opposite way. Actually, $\Delta\gamma_{\text{GPP}}^{\text{Temp}}\gamma_{\text{Temp}}^{\text{ENSO}}$

show the negative value for North America (Fig. 2c), while southern South America has negative in $\Delta Y_{Temp}^{ENSO, Temp} Y_{GPP}^{Temp}$ (Fig. 2e).

Paragraphs beginning at lines 153 and 159: can be combined.

Response: We have merged mentioned two paragraphs into one.

Line 166: This was confusing. The remote impact of ENSO *is* altered under greenhouse warming, as the authors show in the very next paragraph. Do you mean that even *if* the remote impact were not to change, the GPP sensitivity would still enhance ENSO/carbon cycle feedbacks? If so please say that instead.

Response: Yes, even though the remote impact of ENSO, such as temperature and precipitation anomalies to ENSO, is fixed, GPP anomaly to ENSO can be intensified by changing GPP sensitivity to temperature or precipitation (Fig. 2c,d). Actually, the current climate state had already close or exceed the optimum temperature range to GPP that follows the parabolic curve and having optimum range at the center (Yamori et al., 2014), suggesting climatological warming may lead enhanced GPP sensitivity to temperature (Fig. 1 in Niu et al., 2012). We have added more details in the revised manuscript as follows:

(P6 L149-152) “This suggests that the carbon flux of GPP will be more sensitive to local temperature variations under greenhouse warming. This is consistent with previous studies, which suggest that current climate state had already close or exceed the optimal temperature range to terrestrial productivity^{15,16}.”

Lines 198-9: This is an interesting result! I'd just suggest including references for the negative evaporation/soil moisture feedbacks in soil moisture-limited regimes.

Response: We have added more previous studies related to evaporation/soil moisture feedbacks that are Fischer et al. (2007), Jung et al. (2010), and Miralles et al. (2014).

Lines 258-9: Other appropriate references to include on internal ENSO variability and climate change: Wittenberg 2009 GRL, Stevenson et al. 2010, 2012 J. Climate.

Response: We have added that references.

Line 285: Specify explicitly that the partial regression coefficients here are derived from a multiple linear regression of GPP on T, P.

Response: We have added your suggestion into Method section as follows:

(P13 L304-309) “Hence, to estimate the sensitivities of GPP to temperature and precipitation separately, a partial regression was employed in this study on behalf of the partial differential for the sensitivity of the GPP to surface temperature and precipitation and partial regression coefficients are derived from multiple linear regression of GPP on surface temperature and precipitation”

Figure 1:

- Font sizes don't match between panels a/d, b/c/e/f
- Are the regressions plotted here simultaneous in time?
- How many ensemble members are used for each model?
- Panel g: PI, ECP should be plotted in different colors or intensities, right now they're hard to distinguish. Also there should be error bars on the ECPs as well.

Response: Revised as following your suggestion

Figure 2:

- Label inset boxes
- X tick labels for regions need to be easier to understand: suggest adding some kind of legend for decoding the abbreviations

Response: Revised as following your suggestion

Figure 3:

- Suggest eliminating X tick labels on panels a-d, since they get mixed up with the superscripts on the gamma's
- Why are the regression coefficients missing over N. Africa?

Response: We added more space between panels to avoid mixed up.

In N. Africa, most models do not simulate GPP variation because of Sahara Desert.

Figure 4: Switch the sign of the color bar in panels c,d - right now it is confusing.

Response: Revised as following your suggestion

Supp Figure 1:

- Add panel labels (a,b,c)
- The legend is hard to read; make bigger and/or bolder text
- The NBP and GPP colors are nearly indistinguishable, at least when I printed the figure. Make one dashed and/or a different color

Response: We have changed previous Supplementary Fig. 1 (Supplementary Fig. 2; Fig. D4).

Supp Figures 2, 4: Same comments as Figure 2

Response: Revised as following your suggestion

References

- Collins, M. *et al.* The impact of global warming on the tropical Pacific Ocean and El Niño. *Nat. Geosci.* **3**, 391–397 (2010).
- Kim, J. S., Kug, J. S., Yoon, J. H. & Jeong, S. J. Increased atmospheric CO₂ growth rate during El Niño driven by reduced terrestrial productivity in the CMIP5 ESMs. *J. Clim.* **29**, 8783–8805 (2016).
- Kloster, S., Mahowald, N. M., Randerson, J. T. & Lawrence, P. J. The impacts of climate, land use, and demography on fires during the 21st century simulated by CLM-CN. *Biogeosciences* **9**, 509–525 (2012).
- Kloster, S. & Lasslop, G. Historical and future fire occurrence (1850 to 2100) simulated in CMIP5 Earth System Models. *Global Planet. Change* **150**, 58–69 (2017).
- Van der Werf, G. R., Randerson, J. T., Giglio, L., Collatz, G. J., Kasibhatla, P. S. & Arellano, A. F. Interannual variability in global biomass burning emissions from 1997 to 2004. *Atmos. Chem. Phys.*, **6**, 3423–3441 (2006).
- Yamori, W., K., Hikosaka, D. A., Way, Temperature response of photosynthesis in C₃, C₄, and CAM plants: temperature acclimation and temperature adaptation. *Photosynth. Res.* **119**, 101–117 (2014).

- Niu, S, *et al.* Thermal optimality of net ecosystem exchange of carbon dioxide and underlying mechanisms. *New Phytol.* **194**, 775–783 (2012).
- Fischer, E. M., Seneviratne, S. I., Lüthi, D. & Schär, C. Contribution of land–atmosphere coupling to recent European summer heat waves. *Geophys. Res. Lett.* **34**, L06707 (2007).
- Jung, M. *et al.* Recent decline in the global land evapotranspiration trend due to limited moisture supply. *Nature* **467**, 951–954 (2010).
- Miralles, D. G., Teuling, A. J., van Heerwaarden, C. C. & Vila-Guerau de Arellano, J. Mega-heatwave temperatures due to combined soil desiccation and atmospheric heat accumulation. *Nat. Geosci.* **7**, 345–349 (2014).

Figure D1 | NBP sensitivities to ENSO. (a) Regression coefficients of carbon flux anomalies in 40°S–40°N on the DJF Niño3.4 index for NBP based on the pre-industrial experiment (blue), RCP4.5 (red), and (b) the differences between the two experiments. Shaded area and dashed lines indicate 95% confidence levels calculated using bootstrap method. (c) Regression coefficients of SONDJF NBP anomalies in 40°S–40°N on DJF Niño3.4 for the pre-industrial experiment and RCP4.5. Error bars indicate 95% confidence levels of regression coefficients calculated using bootstrap method. * $P < 0.1$ and ** $P < 0.05$ for difference between two experiments in regression coefficients being significantly different from zero.

Figure D2 | Composite maps of mean precipitation (**left**) and temperature anomaly (**right**) for July(0)–June(1) during El Niño years from the Earth System model's MME. (from Kim et al. 2016)

Figure D3 | MME composite maps of CO₂ flux due to NPP anomalies during El Niño events in D(0)JF(1) (from Kim et al. 2016).

Fig. 1 in Niu et al., 2012 | Conceptual figure for the shifts of optimum temperature of net ecosystem productivity (NEP; $NEP = -NEE$ (net ecosystem exchange)) as a result of the changes in optimum temperature of gross primary productivity (GPP) or the temperature sensitivity of respiration. (a) Here it is assumed that in a warmer year or at a warmer site, the optimum temperature of NEP shifts higher owing to a shift of optimum temperature of GPP. In (b) it is assumed that the shifts in the optimum temperature of NEP are the result of the temperature acclimation of Re (decrease of Q_{10}). In (c) it is assumed that the optimum temperature of NEP shifts higher owing to acclimation of both GPP and Re. The dashed curves represent the temperature response curve in a warmer year or at a warmer site. The vertical lines refer to the maximum NEP.

Figure D4 | Terrestrial carbon flux sensitivities to ENSO. Regression coefficient of carbon flux anomalies in 40°S – 40°N on the DJF Niño3.4 index for (a) GPP, (c) R_a (e) R_h and (g) fire in the pre-industrial experiment (blue) and ECP4.5 (red) and the (b, d, f, h) difference between two experiments for MME (thick line) and 95% confidence level (shaded) based on bootstrap estimates.

Reviewer #1 (Remarks to the Author):

The authors have done an excellent job of responding to the substantive review concerns and the paper is ready for acceptance. However, for it to be published, issues of clarity need to be addressed, and particularly in one section.

Line 51: Therefore, exact evaluations of future changes in the relationships between ENSO and the terrestrial carbon cycle are important to understand future variations in the global carbon cycle.

This statement needs to be more explicit about what the authors are looking for.

Line 54 can be more definitive and clearer "The terrestrial carbon cycle will change in response to altered ENSO frequency or intensity in a warmer climate". The text following is confusing and somewhat jargon-y. The authors are implying nonlinear responses to temperature such that excursions may have a bigger carbon impact if the average is higher but they use a lot of jargon and are wordy without actually coming out and stating what they mean. This issue was reviewed in some detail in Wang and Schimel a number of years ago.

Line 83 is inside out "to examine...we..." instead of the simpler "The sensitivity of...was estimated from...".

Line 106. Not processes such as but exactly those fluxes. The processes, growth, decomposition, disturbance, allocation etc are different. So, no "such as" and not processes in this context but "fluxes".

The sentence beginning on line 108 is also inside out "To determine" could be "The sensitivity of ...was estimated from the separate responses of"

The phrase beginning "have negligible values" is incomprehensible. These fluxes can't be negligible, R must nearly equal GPP since NBP is the difference. What is meant here? Anomalies? If true, it implies a major model error.

Line 130, I'd prefer the sentence right side out "The ENSO response was decomposed... to understand..."

Line 140, model results don't suggest, either the model is nonlinear or it's not! Clarify.

The paragraph beginning on line 144 is incomprehensible. It isn't clear what "negative" means as a flux, what is a parameter here? It seems to mean a flux component (GPP, fire etc) but is something else meant? What does "remote" mean? This needs a total rewrite. What does enhanced mean? Larger temperature anomalies during future El Ninos compared to today?

Also, on terminology, I assume this study is actually of El Nino, which seems to be used interchangeably with ENSO. However, the Southern Oscillation also includes the La Nina phase, which from the text, I don't believe the authors examined. The terms should be clear and consistent. I'd prefer the authors use El Nino rather than ENSO, and be clear they examined the warm phase of the ENSO cycle.

The section beginning on line 189 is absolutely crucial. The model carbon cycles may be almost entirely erroneous, as these models differ widely and have relatively low skill. However, this coupling is the heart of the analysis and could well be robust even if the GPP-R-Fire partitioning is incorrect (which is almost certainly true as the author snote). The coupling of carbon, temperature and the water cycle needs to be highlighted in the abstract and is the unqie and significant finding of this study.

Reviewer #3 (Remarks to the Author):

Overall, I think that the authors made appropriate changes in response to my comments. I have no further comments, except that there are some writing errors that occur in some of the new text, which will require a bit of editing to clean up.

Reviewer #4 (Remarks to the Author):

This is an analysis of the CMIP5 earth system models, designed to understand how the carbon cycle response to the El Nino/Southern Oscillation will respond to future climate change. The authors analyze the 9 CMIP5 models which include an active carbon cycle, and demonstrate that the impacts of ENSO are enhanced in the ECP4.5 simulations relative to the pre-industrial due to both increases in the regional temperature responses to ENSO and the response of gross primary productivity to temperature. They conclude that climate change may result in more variable agricultural productivity, with implications for food security.

General comments:

I thought that the authors did a thorough job in responses to the previous round of reviews. I believe that the work is important and interesting, and would be happy for it to be published in Nature Communications after a few more changes have been made to the manuscript. These are mostly clarifications so should not be difficult to address.

Specific comments:

Line 29: *The* El Nino/Southern Oscillation

Line 34: Phrasing is confusing here - I think you mean pre-industrial *in* future projections, not *and* future projections

Line 62: "directly connected to" grammatically incorrect

Lines 67-68: run-on sentence, confusing and grammatically incorrect

Line 85: How exactly have you done the multi-model averaging for the regression coefficients? Simple arithmetic mean?

Line 97: Add symbol Δ γ etc. to Figure 1b caption

Line 99: This should be -1.29 not +1.29 I believe

Line 105: stronger throughout SONDJF? Or some other season?

Line 107: autotrophic

Line 121: Does this mean that there are no fires simulated at all in the other models? Or was the data simply not available through the CMIP5 archive? The answer could be quite important for this analysis.

Line 122: There needs to be some statement here on what the underestimation of fire carbon emissions means for this particular study. Would it follow that these results are in fact a lower limit on the possible changes to carbon impacts of ENSO?

Line 155: Sentence not correct as written (see Figure 2e). Suggest rephrasing to "even if the remote impact of ENSO..." rather than "even though".

Paragraph at line 158: You have not demonstrated a causal link here between land temperature response to ENSO and a negative GPP response to temperature, and so this sentence needs to either be deleted or rephrased.

Line 172: From Figure 3, SE Africa appears to have a larger reduction than Insular SE Asia. This should be mentioned here.

Line 173: On a related note, Supp Fig 7 shows a dramatic increase in SE Africa, which would imply that the change is not always proportional to the present-climate contribution.

Line 215: This statement is not true at all for Australia! There should be a mention/discussion of what is going on in that region here.

Figure 1: define "MME" in caption. Also specify precisely what the bootstrapping is conducted on for your 95% confidence intervals.

Figure 1b, Supp. 2h: define what is meant by red stars in the difference maps. I assume these relate to significance somehow? Not clear from Fig 1 caption whether the stars in panels b, c follow the same convention.

Figure 3: The Southeast Africa box on the inset map is different from the one in Supp Figure 7. Please ensure that both are plotted the same way, and confirm that the results in the two figures in fact do correspond to equivalent regions.

EDITORIAL REQUESTS:

* Please also review the changes in the attached copy of your manuscript and supplementary information, which has been edited for style, and address the comments and queries I have added. Please use the 'track changes' feature in Word to make the process of accepting your manuscript more efficient.

Response: We have edited the manuscript and supplementary information based on the Editor's comment.

* Data availability statements and data citations policy: All Nature Communications manuscripts must include a section titled "Data Availability" at the end of the Methods section or main text (if no Methods).

Response: We have added "Data Availability" at the end of the Methods section as follows:

(P15 L331-334) **"Data availability**

The data that support the findings of this study are all publicly available from their sources. Processed data, products and code produced in this study are available from the corresponding author upon reasonable request."

* Your paper will be accompanied by a two-sentence editor's summary, of between 250-300 characters, when it is published on our homepage. Could you please approve the draft summary below or provide us with a suitably edited version.

The terrestrial carbon cycle is strongly influenced by El Niño Southern Oscillation (ENSO), but how this relationship will change in future is not clear. Here the authors use state-of-the-art models to show that the sensitivity of the carbon cycle to ENSO will increase under future climate change.

Response: Editor's summary is excellent to present this study.

One suggestion is using "El Niño/Southern Oscillation" rather than "El Niño Southern Oscillation".

REVIEWERS' COMMENTS:

Reviewer #1 (Remarks to the Author):

The authors have done an excellent job of responding to the substantive review concerns and the paper is ready for acceptance. However, for it to be published, issues of clarity need to be addressed, and particularly in one section.

Response: We appreciate the Reviewer #1 for encouraging comments. The reviewer's comments were fully incorporated in the revised manuscript. Our responses to the specific comments are as follows:

Line 51: Therefore, exact evaluations of future changes in the relationships between ENSO and the terrestrial carbon cycle are important to understand future variations in the global carbon cycle. This statement needs to be more explicit about what the authors are looking for.

Response: We have edited that sentence as follows:

(P3 L49-51) "Therefore, understanding how the relationships between ENSO and the terrestrial carbon cycle provides a practical implication on future changes in the global carbon cycle."

Line 54 can be more definitive and clearer "The terrestrial carbon cycle will change in response to altered ENSO frequency or intensity in a warmer climate". The text following is confusing and somewhat jargon-y. The authors are implying nonlinear responses to temperature such that excursions may have a bigger carbon impact if the average is higher but they use a lot of jargon and are wordy without actually coming out and stating what they mean. This issue was reviewed in some detail in Wang and Schimel a number of years ago.

Response: We have edited that sentence as following your comment and we have added a citation that Wang and Schimel, 2003.

(P3 L52-53) "The terrestrial carbon cycle will change in response to altered ENSO teleconnection in a warmer climate⁹."

However, in our analysis as the linear regression using Niño3.4 index, it did not take account for changes in ENSO frequency or intensity in the future projection.

Line 83 is inside out "to examine...we..." instead of the simpler "The sensitivity of...was estimated from...".

Response: We have edited that sentence as following your comment.

(P4 L81-83) "The sensitivity of net carbon flux from land (net biome production; NBP) in 40°S–40°N to ENSO ($\gamma_{\text{NBP}}^{\text{ENSO}}$; PgC yr⁻¹ °C⁻¹) was estimated based on the linear regression of NBP anomalies against December–February (DJF) Niño3.4 index."

Line 106. Not processes such as but exactly those fluxes. The processes, growth, decomposition, disturbance, allocation etc are different. So, no "such as" and not processes in this context but "fluxes".

Response: We have edited that sentence as follows:

(P5 L103-105) “The carbon flux of NBP is defined as the summation of carbon fluxes due to gross primary production (total biomass produced by photosynthesis; GPP), autotrophic respiration (R_a), heterotrophic respiration (R_h) and emissions from wild fires in the CMIP5 ESMs.”

The sentence beginning on line 108 is also inside out “To determine” could be “The sensitivity of ... was estimated from the separate responses of”

Response: We have edited that sentence as follows:

(P5 L105-107) “The sensitivities of carbon fluxes to ENSO were estimated from the separate responses of each carbon flux.”

The phrase beginning “have negligible values” is incomprehensible. These fluxes can't be negligible, R must nearly equal GPP since NBP is the difference. What is meant here? Anomalies? If true, it implies a major model error.

Response: We have edited that sentence as follows:

(P5 L108-109) “but carbon fluxes due to respiration, R_a and R_h , have relatively small values for both the experiments and for the differences between the two experiments”

Line 130, I'd prefer the sentence right side out “The ENSO response was decomposed... to understand...”

Response: We have modified that sentence as follows:

(P6 L126-129) “Because it is well-known that the terrestrial GPP is affected by local temperature and precipitation to a large extent (see Methods, Eq. 2)^{7,8,17,18}, the GPP response to ENSO was decomposed into the contributions of local temperature and precipitation to understand how the ENSO-related carbon cycle is intensified under greenhouse warming.”

Line 140, model results don't suggest, either the model is nonlinear or it's not! Clarify.

Response: We analyzed using multiple linear regression. In this sense, GPP response to ENSO in the western Africa cannot be explained by Fig. 2c, d, e, and f. It suggests that residual term in Eq. 4 may contribute to negative GPP response to ENSO.

We have modified that sentence to avoid any confusion as follows:

(P6 L136-138) “In the case of the western Africa, however, the summation of all terms cannot explain the negative value of the changes in the GPP response to ENSO, suggesting that other physical factors might be more important for the changes over the western Africa.”

The paragraph beginning on line 144 is incomprehensible. It isn't clear what “negative” means as a flux, what is a parameter here? It seems to mean a flux component (GPP, fire etc) but is something else meant? What does “remote” mean? This needs a total rewrite. What does enhanced mean? Larger temperature anomalies during future El Ninos compared to today?

Response: As the reviewer suggested, we rewrote this paragraph to clarify our argument.

(P6 L139-153) “For most tropical regions, although it is clear that the contribution of the changes in the GPP sensitivity to temperature ($\Delta\gamma_{GPP}^{Temp, ENSO}$, Fig. 2c and Supplementary Fig. 5) and that of changes in the temperature response to ENSO ($\Delta\gamma_{Temp}^{ENSO, GPP}$, Fig. 2e and Supplementary Fig. 6) exhibits negative in most of the

tropics, such as in Amazonia, Australia, and Insular Southeast Asia. This indicates that they make significant contributions to the changes in the GPP response, the other parameters make relatively small contributions related to precipitation²³. As shown in Fig. 2c, the changes in the GPP sensitivity to temperature ($\Delta\gamma_{GPP}^{Temp} \gamma_{Temp}^{ENSO}$) also make a substantial contribution to the changes in the enhanced GPP response to ENSO ($\Delta\gamma_{GPP}^{ENSO}$) throughout the tropics (Fig. 2c). This suggests that the carbon flux of GPP will be more sensitive to local temperature variations under greenhouse warming. This is consistent with previous studies, which suggest that current climate state had already close or exceed the optimal temperature range to terrestrial productivity^{16,17}. Thus, even if the remote impacts of ENSO on regional climate are not altered under greenhouse warming, the ENSO-related carbon cycle can be intensified owing to changes in GPP sensitivity to temperature ($\Delta\gamma_{GPP}^{Temp}$).”

Also, on terminology, I assume this study is actually of El Nino, which seems to be used interchangeably with ENSO. However, the Southern Oscillation also includes the La Nina phase, which from the text, I don't believe the authors examined. The terms should be clear and consistent. I'd prefer the authors use El Nino rather than ENSO, and be clear they examined the warm phase of the ENSO cycle.

Response: Because we used linear regression against December–February (DJF) Niño3.4 index, all results show enhance impact on carbon cycle in both cases of positive and negative phase of ENSO such as El Nino and La Nina.

The section beginning on line 189 is absolutely crucial. The model carbon cycles may be almost entirely erroneous, as these models differ widely and have relatively low skill. However, this coupling is the heart of the analysis and could well be robust even if the GPP-R-Fire partitioning is incorrect (which is almost certainly true as the author's note). The coupling of carbon, temperature and the water cycle needs to be highlighted in the abstract and is the unique and significant finding of this study.

Response: In the abstract, we already mentioned that “Our findings suggest that the ENSO-related carbon cycle, a dominant natural variability, will be enhanced by hydroclimate changes caused by anthropogenic forcing.”.

Reference

Wang, G. & Schimel, D. Climate change, climate modes, and climate impacts. *Annu. Rev. Environ. Resour.*, **28**, 1–28 (2003).

Reviewer #3 (Remarks to the Author):

Overall, I think that the authors made appropriate changes in response to my comments. I have no further comments, except that there are some writing errors that occur in some of the new text, which will require a bit of editing to clean up.

Response: We appreciate the Reviewer #3's comments. We have edited manuscript by English language editing service.

Reviewer #4 (Remarks to the Author):

This is an analysis of the CMIP5 earth system models, designed to understand how the carbon cycle response to the El Nino/Southern Oscillation will respond to future climate change. The authors analyze the 9 CMIP5 models which include an active carbon cycle, and demonstrate that the impacts of ENSO are enhanced in the ECP4.5 simulations relative to the pre-industrial due to both increases in the regional temperature responses to ENSO and the response of gross primary productivity to temperature. They conclude that climate change may result in more variable agricultural productivity, with implications for food security.

General comments:

I thought that the authors did a thorough job in responses to the previous round of reviews. I believe that the work is important and interesting, and would be happy for it to be published in Nature Communications after a few more changes have been made to the manuscript. These are mostly clarifications so should not be difficult to address.

Response: We thank the Reviewer #4 for his/her time and critical comments. Our responses to the specific comments are as follows:

Specific comments:

Line 29: *The* El Nino/Southern Oscillation

Response: Corrected.

Line 34: Phrasing is confusing here - I think you mean pre-industrial *in* future projections, not *and* future projections

Response: Corrected.

Line 62: "directly connected to" grammatically incorrect

Response: We have modified that sentence as follows:

(P3 L59-60) "thereby changes in sensitivity depending on varying climate states would be directly related to the strength of the terrestrial response to ENSO."

Lines 67-68: run-on sentence, confusing and grammatically incorrect

Response: We have modified that sentence as follows:

(P3 L64-69) "In this study, we examine future changes in the sensitivity of the terrestrial carbon cycle to ENSO in the Coupled Model Intercomparison Project Phase 5 (CMIP5) Earth System Models (ESMs) by comparing pre-industrial and future projections. We analyzed Extended Concentration Pathway 4.5 (ECP4.5) scenario for the future projections which is the extension of the Representative Concentration Pathways (RCP) 4.5 with continuous anthropogenic forcing until 2300."

Line 85: How exactly have you done the multi-model averaging for the regression coefficients? Simple arithmetic mean?

Response: We did simple arithmetic mean for MME. We have added this information into Method section as follows:

(P13 L271-272) “The MME for the regression coefficient is defined that simple arithmetic mean of nine ESMs.”

Line 97: Add symbol Δ γ etc. to Figure 1b caption

Response: We have added that symbol into Figure 1b caption.

Line 99: This should be -1.29 not +1.29 I believe

Response: Corrected.

Line 105: stronger throughout SONDJF? Or some other season?

Response: We have edited as follows:

(P5 L100-102) “In particular, among nine ESMs, six (eight) models exhibit significantly stronger sensitivities of NBP in SONDJF at the 95% (90%) confidence level.”

Line 107: autotrophic

Response: Corrected.

Line 121: Does this mean that there are no fires simulated at all in the other models? Or was the data simply not available through the CMIP5 archive? The answer could be quite important for this analysis.

Response: Only four models have the simulation of carbon flux due to nature fire, so we have edited that sentence as follows:

(P6 L116-118) “only four models, CESM1-CAM5, IPSL-CM5A-MR, MPI-ESM-LR and NORESM1-M, which have the simulation of carbon flux due to nature fire”

Line 122: There needs to be some statement here on what the underestimation of fire carbon emissions means for this particular study. Would it follow that these results are in fact a lower limit on the possible changes to carbon impacts of ENSO?

Response: As we described above, NBP in five models, which does not have simulation of fire, do not include carbon flux due to fire. Because of that, the CMIP5 ESMs tend to underestimate NBP anomalies due to a lack of the fire simulation. We have modified mentioned statement as follows:

(P6 L115-119) “Even though this MME result is equal to 20% of GPP anomalies in SONDJF, it is based on only four models, CESM1-CAM5, IPSL-CM5A-MR, MPI-ESM-LR and NORESM1-M, which have the simulation of carbon flux due to nature fire. Because other models do not have carbon emission anomalies by fire, the CMIP5 ESMs tend to underestimate of the fire contribution on NBP anomalies.”

Line 155: Sentence not correct as written (see Figure 2e). Suggest rephrasing to "even if the remote impact of ENSO..." rather than "even though".

Response: Corrected.

Paragraph at line 158: You have not demonstrated a causal link here between land temperature response to ENSO and a negative GPP response to temperature, and so this sentence needs to either be deleted or rephrased.

Response: We have edited that paragraph as follows:

(P7 L154-160) “In addition, Figure 2e, $\Delta Y_{Temp}^{ENSO} / \Delta Y_{GPP}^{Temp}$, shows a contribution of changes

in the land temperature response to ENSO ($\gamma_{\text{Temp}}^{\text{ENSO}}$) on changes in GPP response to ENSO. Because the GPP response to local temperature ($\gamma_{\text{GPP}}^{\text{Temp}}$) is mostly negative in tropical regions, enhanced $\gamma_{\text{Temp}}^{\text{ENSO}}$ under greenhouse warming lead to negative GPP response to ENSO. In particular, the enhanced temperature response to ENSO is clearly found over the Amazon and eastern Africa. The enhanced temperature response during El Niño tends to reduce the carbon uptake by GPP owing to enhanced heat stress to ecosystems.”

Line 172: From Figure 3, SE Africa appears to have a larger reduction than Insular SE Asia. This should be mentioned here.

Response: We have mentioned SE Africa as follows:

(P7 L168-170) “Of these regions, Amazonia, Australia, Insular Southeast Asia and Southeast Africa considerably contribute to the reduced carbon fluxes out of atmosphere due to GPP on land, i.e., -0.31 , -0.18 , -0.13 and -0.14 $\text{PgC yr}^{-1} \text{ } ^\circ\text{C}^{-1}$, respectively (Fig. 3).”

Line 173: On a related note, Supp Fig 7 shows a dramatic increase in SE Africa, which would imply that the change is not always proportional to the present-climate contribution.

Response: We have edited that sentence as follows:

(P8 L170-171) “This regional contribution appears to be proportional to their present-climate contribution, except for Southeast Africa (Supplementary Fig. 7).”

Line 215: This statement is not true at all for Australia! There should be a mention/discussion of what is going on in that region here.

Response: In Australia, future changes in soil moisture are weak and correlation between evaporation and temperature also does not show differences between two experiments (Figs. 4b and 4d). Therefore, we can conclude that the spatial pattern of future changes in the evaporation–temperature coupling strength is similar to future changes in soil moisture in terms of weak signals in Australia.

Figure 1: define "MME" in caption. Also specify precisely what the bootstrapping is conducted on for your 95% confidence intervals.

Response: We have added that information into Method section as follows:

(P13 L271-276) “The MME for the regression coefficient is defined that simple arithmetic mean of nine ESMs. Bootstrap resampling is used to assess the significance of MME. 10000 bootstrap samples are produced for the regression coefficient by selecting nine ESMs randomly with replacement. These are used to produce 10000 estimates of the MME, which constitute an empirical bootstrap distribution, allowing a confidence interval for the MME.”

Figure 1b, Supp. 2h: define what is meant by red stars in the difference maps. I assume these relate to significance somehow? Not clear from Fig 1 caption whether the stars in panels b, c follow the same convention.

Response: We have added that information as follows:

(P24 L505-506) “Red stars indicate significant months for $\Delta\gamma_{NBP}^{ENSO}$ at 95% confidence levels.”

Figure 3: The Southeast Africa box on the inset map is different from the one in Supp Figure 7. Please ensure that both are plotted the same way, and confirm that the results in the two figures in fact do correspond to equivalent regions.

Response: Those are same, but it seems that there is a problem during converting to PDF file. Now we have made PDF file by ourselves. Please check SI file.